# Gasdermin D cysteine residues synergistically control its palmitoylation-mediated membrane targeting and assembly

Eleonora Margheritis [ID][1], Shirin Kappelhoff [ID][1], John Danial[2,3,4], Nadine Gehle [ID][1], Wladislaw Kohl[1], Rainer Kurre [ID][1], Ayelén González Montoro[1] & Katia Cosentino [ID][1][✉]

## Abstract

**Gasdermin D (GSDMD) executes the cell death program of pyroptosis by assembling into oligomers that permeabilize the plasma membrane. Here, by single-molecule imaging, we elucidate the yet unclear mechanism of Gasdermin D pore assembly and the role of cysteine residues in GSDMD oligomerization. We show that GSDMD preassembles at the membrane into dimeric and trimeric building blocks that can either be inserted into the membrane, or further assemble into higher-order oligomers prior to insertion into the membrane. The GSDMD residues Cys39, Cys57, and Cys192 are the only relevant cysteines involved in GSDMD oligomerization. S-palmitoylation of Cys192, combined with the presence of negatively-charged lipids, controls GSDMD membrane targeting. Simultaneous Cys39/57/192-to-alanine (Ala) mutations, but not Ala mutations of Cys192 or the Cys39/57 pair individually, completely abolish GSDMD insertion into artificial membranes as well as into the plasma membrane. Finally, either Cys192 or the Cys39/Cys57 pair are sufficient to enable formation of GSDMD dimers/trimers, but they are all required for functional higher-order oligomer formation. Overall, our study unveils a cooperative role of Cys192 palmitoylation-mediated membrane binding and Cys39/57/192-mediated oligomerization in GSDMD pore assembly. This study supports a model in which Gasdermin D oligomerization relies on a two-step mechanism mediated by specific cysteine residues.**

**Keywords** Gasdermins; Pyroptosis; Stoichiometry; Cysteine-mediated Oligomerization; S-palmitoylation
**Subject Categories** Autophagy & Cell Death; Membranes & Trafficking; Post-translational Modifications & Proteolysis

## Introduction

Pyroptosis is a highly inflammatory cell death program that plays a crucial role in the host's response to infected or damaged cells (Broz and Dixit, 2016). Gasdermin (GSDM) proteins are central executors of pyroptosis by forming pores on the plasma membrane (PM). These pores mediate the secretion of inflammatory cytokines to recruit immune cells and, in certain conditions, induce cell lysis (Broz et al, 2020; Feng et al, 2018; Liu et al, 2021). Excessive pyroptosis is associated with severe inflammatory and autoimmune diseases (Li et al, 2020; Liu et al, 2021). Therefore, tight regulation of this process is critical for controlling inflammation in order to eliminate infected and injured cells while preventing exacerbation of inflammation.

GSDMs are initially expressed as harmless cytosolic proteins and only become active after protease-mediated cleavage of their cytotoxic N-terminal domain (NTD) from their autoinhibitory C-terminal domain (CTD) (Broz et al, 2020; He et al, 2015; Kayagaki et al, 2015; Shi et al, 2015). GSDMD, the most comprehensively characterized member of the family, can be activated by cleavage by several proteases, including the inflammatory caspases 1 and 4/5 (11 in mice), corresponding to the canonical and non-canonical pathways of inflammasome activation, respectively (Kayagaki et al, 2015; Liu et al, 2020; Shi et al, 2015). Once the GSDMD NTD is cleaved, it translocates to the inner leaflet of the PM where it assembles into oligomers that eventually cause pore formation (Aglietti et al, 2016; Ding et al, 2016; Kappelhoff et al, 2024; Liu et al, 2016; Sborgi et al, 2016).

At a mechanistic level, the exact details that determine the recruitment of GSDMD to the membrane, the process of oligomerization, and membrane permeabilization remain elusive. Structural analysis of GSDMD in its soluble and pore-forming configurations has highlighted profound conformational changes of the NTD that enable membrane insertion and perforation (Ding et al, 2016; Liu et al, 2019; Xia et al, 2021). In the cryo-EM pore structure, GSDMD assembles a β-barrel pore with a 33-fold symmetry and a pore size of 21 nm (Xia et al, 2021). In the same study, a fully assembled prepore ring structure in which complete oligomerization occurred without membrane insertion was identified. This suggests that oligomerization

[1]Department of Biology/Chemistry and Center for Cellular Nanoanalytics (CellNanOs), University of Osnabrück, Osnabrück, Germany. [2]Yusuf Hamied Department of Chemistry, University of Cambridge, Cambridge, UK. [3]UK Dementia Research Institute, University of Cambridge, Cambridge, UK. [4]Present address: School of Physics and Astronomy, University of St Andrews, North Haugh, St Andrews, UK. [✉]E-mail: katia.cosentino@uni-osnabrueck.de

occurs at the membrane prior to the conformational reorganization required for membrane insertion (Xia et al, 2021). In contrast, dynamic atomic force microscopy (AFM) studies in artificial membranes revealed heterogeneous membrane-inserted GSDMD structures, such as arcs and slits, which can evolve into rings over time and are all equally capable of forming pores of different sizes up to 30 nm (Mulvihill et al, 2018; Sborgi et al, 2016). These data, together with experimental observations of ion flux during early pyroptosis (de Vasconcelos et al, 2019) and recent molecular dynamics (MD) simulations indicating pore functionality of small GSDMD oligomers (Schaefer and Hummer, 2022), support an alternative model for pore assembly in which small GSDMD intermediates may already penetrate the membrane and grow into fully assembled ring structures. Whether GSDMD inserts the membrane as a monomer or has to preassemble into a minimal oligomeric unit before insertion and whether small GSDMD oligomers are already able to form functional pores remains to be clarified.

Membrane recruitment of GSDMD is strictly dependent on lipid composition, namely negatively charged lipids and in particular phosphatidylinositol phosphates (PIPs) such as Phosphatidylinositol 4,5-bisphosphate (PI(4,5)$P_2$) (Ding et al, 2016; Mulvihill et al, 2018; Santa Cruz Garcia et al, 2022). However, it is not clear whether other factors, e.g., post-translational modifications (PTMs), also regulate GSDMD targeting to the PM. Interestingly, the pore-forming activity of the GSDME family member is enhanced by palmitoylation of a cysteine in its CTD (Hu et al, 2020b), a PTM that has its evolutionary roots in bacterial GSDMs (Johnson et al, 2022). Whether this is a conserved mechanism in the GSDM family requires further investigation.

Cysteines indeed play a crucial role in GSDMD activity (Devant et al, 2023; Kappelhoff et al, 2024; Liu et al, 2016; Rathkey et al, 2018). Mutations of mouse GSDMD (mGSDMD) Cys39 and especially Cys192 impair oligomerization (Liu et al, 2016). Moreover, targeting of Cys191/Cys192 (human and mouse, respectively) by Cys-reactive inhibitors prevents GSDMD-mediated pyroptosis (Hu et al, 2020a; Humphries et al, 2020; Rathkey et al, 2018). Since reducing conditions break down GSDMD oligomers, it was initially assumed that cysteines mediate oligomerization through the formation of intra- and inter-disulfide bonds (Liu et al, 2016). However, in the solved soluble crystal structure (resolved under reducing conditions) (Liu et al, 2019) and in the membrane-inserted structure (Xia et al, 2021), cysteines are not close enough to each other, which calls into question their role in the formation of disulfide bonds during the oligomerization process. Nevertheless, a cellular redox environment controls GSDMD oligomerization in a cysteine-dependent manner by a mechanism that remains to be elucidated (Devant et al, 2023; Evavold et al, 2021). Interestingly, although mutation of Cys191/Cys192 is sufficient to impair GSDMD functionality, oxidation occurs at multiple cysteines (Devant et al, 2023; Wang et al, 2019). In addition to oxidation, other PTMs are known to occur on cysteines, including itaconation of Cys77 and succination of Cys56/191/268/309/467 (human) (Bambouskova et al, 2021; Humphries et al, 2020), both of which inhibit pore-forming activity. Taken together, these data indicate that the modification of cysteines is a conserved mechanism for regulating GSDMD activity. Accordingly, determining the specific role of the different cysteines would lead to a better understanding of the detailed mechanism of GSDMD pore formation.

To unravel the mechanism of assembly of GSDMD oligomers in membranes and to reveal the specific contribution of individual cysteines to GSDMD pore formation, we used single-molecule imaging (SMI) microscopy combined with intensity-based stoichiometry analysis. SMI is a powerful technique for visualizing weak or rare events, such as single fluorescent protein units, and, in combination with stoichiometry, for identifying individual oligomerization states, which would be lost in ensemble measurements (Aggarwal and Ha, 2016; Hallworth and Nichols, 2012; Jenner et al, 2020; Margheritis et al, 2023; Ulbrich and Isacoff, 2007). With this approach, we show that GSDMD preassembles at the membrane into dimeric and trimeric building blocks from which higher-order oligomers can be generated. Oligomerization can precede membrane insertion. Once dimers and trimers are formed, membrane insertion can occur at any step of the oligomerization process. We found that the targeting of GSDMD to the membrane is controlled by cysteine-mediated S-palmitoylation at Cys192 in combination with the presence of negatively charged lipids, namely PI(4,5)$P_2$. Intriguingly, only combined C39A-C57A-C192A mutations, but not single C192A or C39A-C57A mutations, fully abolish the insertion of GSDMD into lipid bilayers. This effect could also be recapitulated in cells, thanks to a technology called polymer-supported plasma membranes (PSPMs), which allows GSDMD oligomers to be unambiguously identified at the PM of pyroptotic cells (preprint: Kappelhoff et al, 2023). The inactive C192A or C39A-C57A variants individually retain the ability to bind membranes and can assemble into dimers and trimers, suggesting redundancy of these cysteines for dimer/trimer formation. However, they are all required to form higher-order oligomers. This explains the inability of these mutants to permeabilize membranes in liposomes and to promote pyroptosis in cells. Altogether, our quantitative biophysical analysis of GSDMD assembly reveals a cooperative function of Cys39, Cys57, and Cys192 in controlling GSDMD membrane targeting, oligomerization, and membrane insertion. We propose a model in which Cys192 has a dual role: stabilizing membrane binding via palmitoylation, thereby increasing local protein concentration at the membrane, and promoting the formation of GSDMD oligomers. This oligomerization function is shared with Cys39 and Cys57 and involves a two-step mechanism based on transient thiol-mediated interactions for dimer/trimer formation and subsequent higher-order oligomerization. Our work provides conclusive evidence for the regulatory role of these key cysteines in GSDMD pore formation.

## Results

### Dimers and trimers are the basic building blocks of GSDMD oligomerization

To elucidate the molecular steps of GSDMD pore assembly in membranes, we performed stoichiometry analysis of mGSDMD oligomers by single-molecule total internal reflection fluorescence (TIRF) microscopy as in (Cosentino et al, 2022; Danial et al, 2022). This approach involves the reconstitution of protein complexes in a controlled, minimalist membrane environment, thereby excluding regulatory and unknown cellular mechanisms. To this end, we produced a recombinant version of mouse GSDMD fused to mEGFP located immediately upstream of the Caspase 11-mediated cleavage site (mGSDMD-mEGFP) (Fig. 1A). This construct retains full GSDMD activity (Rathkey et al, 2017) as verified by liposome

leakage assay (Appendix Fig. S1A,B), and modeling based on the GSDMD pore structure showed that adjacent mEGFP units are sufficiently flexible and distant from each other to avoid quenching or steric hindrance (Fig. 1B). We first ensured that the purified full-length protein was monomeric and that cleavage by Caspase 11 did not alter the oligomeric state of the protein in solution (Fig. EV1). SDS and native PAGE immunoblot, mass photometry, and stoichiometric analysis of mGSDMD-mEGFP captured on a functionalized glass slide confirmed that recombinant GSDMD appeared largely as a monomer in solution (Fig. EV1A–F). A certain percentage of dimers, which varied between purification batches (Fig. EV1F), was also detected. As these dimers could not be cleaved by Caspase 11 (Fig. EV1G), they do not participate in the oligomerization process and, in agreement with other studies (Kopp et al, 2023; Liu et al, 2016), we attributed them to purification artifacts not relevant for our further analysis. To reconstitute mGSDMD-mEGFP oligomers, we incubated activated GSDMD with liposomes containing 1% of the negatively charged lipid PI(4,5)P₂, which is known to favor GSDMD pore activity (Ding et al, 2016; Mulvihill et al, 2018). After one hour of incubation, the resulting proteoliposomes were fused on a glass slide to obtain GSDMD-containing supported lipid bilayers (SLBs) for TIRF imaging (Fig. 1A). The mGSDMD-mEGFP oligomers appeared as bright spots (Fig. 1C), whose density on the membrane increased with protein concentration (Fig. 1D,E). Similarly, brightness analysis of mGSDMD-mEGFP oligomers showed a narrow distribution that broadened in a concentration-dependent manner (Fig. 1F,G), indicating the formation of high-order oligomers. We then estimated the stoichiometry of the mGSDMD-mEGFP oligomers based on the average intensity of the recombinant mEGFP monomers using our stoichiometry analysis software SAS (Figs. 1C and EV1H–J, see Methods and (Danial et al, 2022)). Stoichiometry analysis, in steady-state conditions, revealed the presence of dimeric and trimeric, but not monomeric, units (Fig. 1H). Interestingly, increasing concentrations of GSDMD led to the formation of higher-order oligomers, variable in proportion between the single experiments, but all preferentially resulting from the combination of dimers and trimers, rather than from the sequential addition of monomers (Fig. 1I). Furthermore, reducing the incubation time from one hour to 5 and 2 min did not alter the pattern of detected oligomeric species and did not disclose inserted monomers both at low and high protein concentrations (Appendix Fig. S1C–F). These data unequivocally indicate that the assembly of GSDMD is initiated by the formation of dimeric and trimeric building blocks from which higher-order oligomers are derived.

## GSDMD oligomerization can precede membrane insertion

Previous results reported that in this experimental setup, membrane proteins spanning the lipid bilayer interact non-specifically with the glass support preventing protein diffusion within the membrane and thus further oligomerization (see (Subburaj et al, 2015) and (Cosentino et al, 2022)). Therefore, we devised to use this as a strategy to investigate whether GSDMD oligomerization following membrane association precedes or not membrane insertion. To this end, we added mGSDMD-mEGFP directly to preformed PI(4,5)P₂-containing SLBs (Fig. 2A). In contrast to the

reconstitution of GSDMD oligomers in liposomes (Fig. 1A), which are free-standing bilayers that allow free diffusion and thus oligomerization of membrane-inserted proteins, here we expected that, if membrane insertion precedes oligomerization, addition of mGSDMD-mEGFP to the SLB would result in immobile monomers due to protein interactions with the glass support. First, we verified that Caspase 11-cleaved mGSDMD-mEGFP, but not full-length mGSDMD-mEGFP, was able to bind to the SLB. Full-length mGSDMD-mEGFP retained only minimal binding (Fig. 2B,C). We then performed a tracking analysis of activated mGSDMD-mEGFP after one hour of incubation on SLB and observed that the particles were mostly immobile (Fig. 2D and Movie EV1). Unexpectedly, however, brightness analysis showed a broad intensity distribution (Fig. 2E), similar to that in proteoliposomes (Fig. 1G), suggesting the presence of oligomers. Stoichiometric analysis at protein concentrations ranging from 0.5 nM to 2 nM confirmed the presence of high-order oligomers, rigorously as combinations of dimers and trimers, with 5-, 6-, and 9-mers as predominant species (Fig. 2F; Appendix Fig. S2A,B), in complete accordance to the previous experiments (Fig. 1I). We also analyzed the stoichiometry of the few particles detected in the mGSDMD-mEGFP full-length sample (Fig. 2B). Although the reduced number of detected particles compared to the active protein limits the robustness of these results, we observed a prevalence of dimeric species (Appendix Fig. S2C). These species may resemble purification-derived dimers (Fig. EV1) or result from aggregated monomeric full-length protein. Regardless, the difference in density to the active protein (Fig. 2C), allows us to exclude the possibility that these non-cleaved oligomers could falsify our stoichiometry analysis.

These results clearly indicate that, in this experimental setting, the assembly occurs prior to insertion into the membrane. Importantly, we observed not only one type but multiple oligomeric species coexisting in the bilayer. This indicates that dimers and trimers, once assembled, can either directly insert into the membrane or further oligomerize into higher-order oligomers ahead of membrane insertion. Thus, our data suggest that, once dimers and trimers are formed, insertion into the membrane can occur at any step in the oligomerization process (Fig. 2F).

## GSDMD membrane binding is impaired in a reducing environment

Several studies have reported that a change in the cellular redox environment plays an important regulatory role in GSDMD function (Devant et al, 2023; Evavold et al, 2021; Liu et al, 2016; Wang et al, 2019). Therefore, we tested the effects of the addition of oxidizing or reducing agents on the permeabilization activity, membrane binding, and oligomerization of GSDMD. Treatment of GSDMD with hydrogen peroxide (H₂O₂) as an oxidizing agent had no effect on the overall GSDMD-mediated liposome leakage or on the kinetics of permeabilization (Fig. EV2A,B), and (Du et al, 2024). Furthermore, stoichiometry analysis revealed that the presence of an oxidative environment did not lead to a shift in the distribution of GSDMD oligomers towards higher-order oligomers, as expected based on previous findings (Devant et al, 2023; Liu et al, 2016; Wang et al, 2019) (Fig. EV2C–F and compare with Fig. 1F–I). We therefore concluded that oxidation does not play a direct role in GSDMD assembly. However, we cannot exclude that it favors the

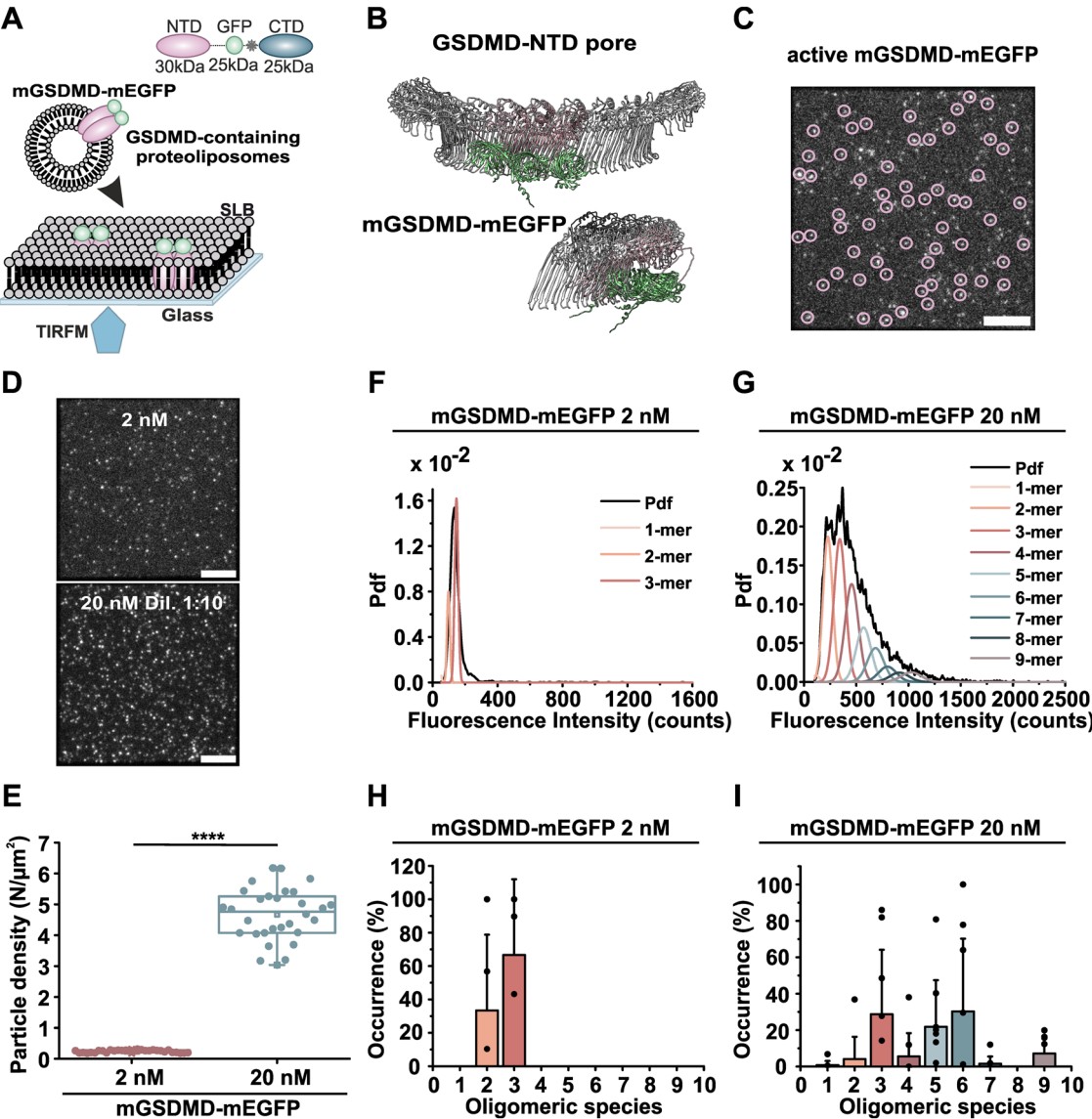

**Figure 1. Oligomerization of GSDMD proceeds by sequential addition of dimers and trimers.**

(A) Schematic representation of GSDMD protein with in pink the N-terminal domain, in green mEGFP, in blue the C-terminal domain, and in gray the caspase cleavage site (up). Scheme of the experimental setup used for stoichiometry experiments in mimetic membranes (bottom). mGSDMD(NTD)-mEGFP oligomers (depicted in pink and green) are reconstituted into LUVs (Egg PC: PI(4, 5)P$_2$ 99:1 mol %) and used to form a supported lipid bilayer (SLB) on glass microscopy chips. (B) Overlay of the prediction for a small oligomer (trimer) of mGSDMD(NTD)-mEGFP run by AlphaFold and the cryo-EM structure of GSDMD pore (pdb 6VF6) (Xia et al, 2021). (C) Representative TIRF image of an SLB containing GSDMD oligomers (bright spots circled in pink). Scale bar 5 μm. (D) Representative TIRF images of SLB-containing GSDMD oligomers prepared from proteoliposomes incubated for 1 h with two different protein concentrations: 2 nM and 20 nM. The sample at 20 nM was diluted 1:10 before SLB preparation to allow single-molecule imaging. Scale bars 5 μm. (E) Particle density of mGSDMD-mEGFP assemblies detected as shown in (D). The values for the 20 nM protein concentration have been corrected for the dilution factor of 10. Plotted are recordings from three independent experiments, ten movies per experiment were analyzed ($p < 0.0001$). Data are presented as box plots with the center line at the median, lower bound at 25th percentile, upper bound at 75th percentile, and whiskers at minimum and maximum values. (F, G) Representative fluorescence intensity distributions of mGSDMD-mEGFP oligomers obtained from samples incubated with 2 nM (F) or 20 nM (G) protein. The obtained brightness distribution was plotted as a probability density function (Pdf, black) and it was fitted with a mixture of Gaussians representing the possible oligomeric states the protein can assume (color). (H, I) Percentage of occurrence of GSDMD oligomeric species (the color code used for the occurrence graphs is the same as for the distributions in (F) and (G)) calculated as the average value from at least eight different experiments (minimum 1500 particles per experiment). Individual experimental data points are indicated as scatter plots in the graphs (0 values are not indicated). Data are corrected for GFP partial labeling (see Methods). Error bars correspond to the SD from the different experiments. Statistics were measured by Student's t-tests with **** for $p < 0.0001$. Source data are available online for this figure.

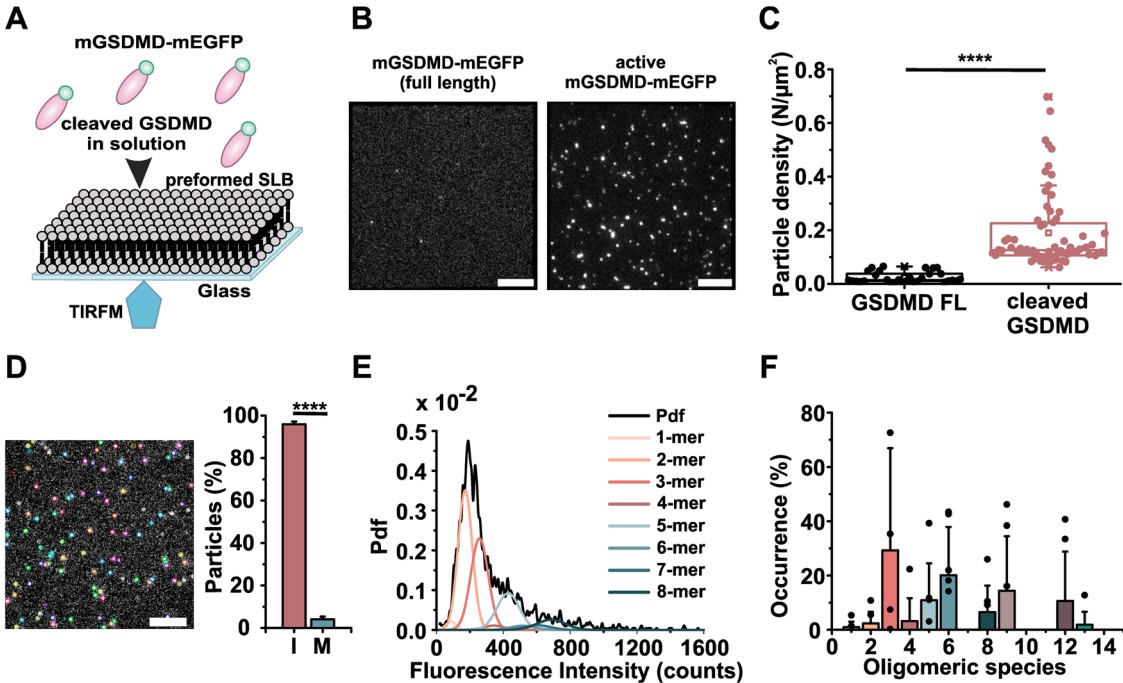

**Figure 2.  GSDMD oligomerization precedes membrane insertion.**

(**A**) Schematic representation of the system used for sample preparation of stoichiometry experiments in model membranes to determine membrane insertion (mGSDMD(NTD)-mEGFP depicted in pink and green). (**B**) Representative TIRF images of an SLB incubated with 2 nM of full-length mGSDMD-mEGFP (left) or cleaved mGSDMD-mEGFP (right) for 1 h. Scale bars 5 µm. (**C**) Density of detected assemblies on SLB for full-length mGSDMD-mEGFP (GSDMD FL) and active mGSDMD-mEGFP (cleaved GSDMD) at 2 nM. Plotted are values obtained from recordings from three independent experiments with at least ten movies per experiment analyzed ($p = 6.7 \times 10^{-9}$). Data are presented as box plots with the center line at the median, lower bound at 25th percentile, upper bound at 75th percentile, and whiskers at minimum and maximum values. (**D**) Particle localization analysis by SLIMfast 4C. Right: representative image of immobile particles detected in the SLB sample and highlighted by colored ROIs (scale bar 5 µm). Left: Percentage of immobile (I) versus mobile (M) particles detected in the sample. Averages from three independent experiments with a minimum of 10 recordings analyzed per experiment ($p < 0.0001$). (**E**) Representative fluorescence distribution of mGSDMD-mEGFP oligomers obtained from preformed SLB incubated with 2 nM mGSDMD-mEGFP activated with 20 nM Caspase 11. The resulting brightness distribution was plotted as a probability distribution function (Pdf, black) and fitted with a mixture of Gaussians to estimate the percentage of occurrence of particles containing n-mer oligomers (color). (**F**) Percentage of occurrence of the different oligomeric species of GSDMD (the color code used for the occurrence graph is the same as for the distributions in (**E**)) calculated as the average from seven independent experiments (minimum 2000 particles analyzed per experiment). Data are corrected for GFP partial labeling. Individual experimental data points are indicated as scatter plots in the graph (0 values are not indicated). Error bars correspond to the SD from the different experiments. Statistics were measured by Student's t-tests with **** for $p < 0.0001$. Source data are available online for this figure.

earlier step of dimer and trimer formation, which was not addressed in our assays. If this were the case, the addition of reducing agents should impair the formation of GSDMD dimers and trimers and lead to the detection of only monomers in our stoichiometry assay. Intriguingly, in the presence of the reducing agent dithiothreitol (DTT), GSDMD permeabilization activity decreased (Fig. EV2G). Unexpectedly, however, our more direct single-molecule imaging analysis, showed a significant reduction in GSDMD signal at the membrane in the presence of different reducing agents, pointing out that GSDMD membrane targeting is compromised under reducing conditions (Fig. 3A,B). This indicates that a redox environment may control the binding of GSDMD to the membrane.

## S-palmitoylation of GSDMD mediates its targeting to membranes but requires the presence of PI(4,5)P$_2$

A recent study has shown that bacterial GSDMs are palmitoylated by an autocatalytic mechanism (Johnson et al, 2022). Palmitoylation or S-acylation is a reversible PTM in which long-chain fatty

acids are attached to cysteine residues via a thioester bond. This modification increases the hydrophobicity of the protein and can regulate protein targeting to membranes and protein-protein interactions, among other processes (Ko and Dixon, 2018). The reversibility of palmitoylation can be influenced by a redox environment, and reducing agents can break the thioester bond in vitro (Ji et al, 2013). Since our assay showed that reducing agents prevent the association of GSDMD with the membrane, this prompted us to test whether our recombinant protein was palmitoylated. To this end, we performed an acyl-biotin exchange (ABE) assay, a method based on the substitution of palmitate groups, removed by hydroxylamine (HA), with Biotin-HPDP and subsequent pull-down of the biotinylated protein using streptavidin beads (Roth et al, 2006a; Roth et al, 2006b). Since free cysteines were preventively blocked with N-ethylmaleimide (NEM), only palmitoylated proteins should be biotinylated (Fig. 3C). Immunoblot analysis showed that HA-treated full-length mGSDMD was highly palmitoylated (Fig. 3D). Consistent with this, particle density analysis by SMI of activated mGSDMD pre-treated with HA showed complete impairment of membrane binding (Fig. 3A,B),

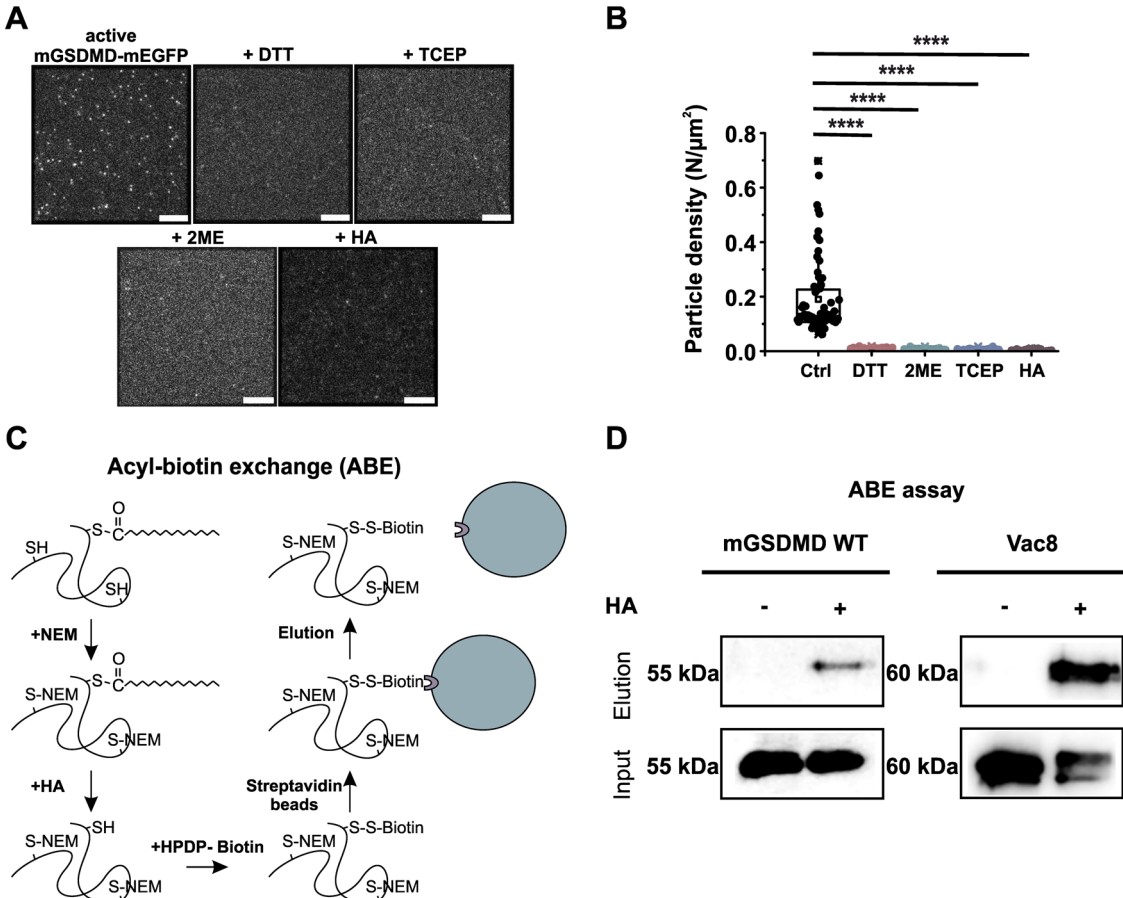

**Figure 3. S-palmitoylation of GSDMD mediates its targeting to membranes.**

(A) Representative TIRF images of SLB prepared from LUVs incubated with active mGSDMD-mEGFP (2 nM) alone (Ctrl) or in the presence of different reducing agents DTT, TCEP, 2ME (all at 1 mM), and 1 M HA showing membrane interaction impairment after treatment. Scale bars 5 μm. (B) Particle density analysis of the number of active mGSDMD-mEGFP assemblies detected in at least two sets of experiments after treatment as in (A) (Ctrl: $n = 60$; DTT: $n = 40$, $p = 1.9 \times 10^{-11}$; 2ME: $n = 30$, $p = 6.3 \times 10^{-9}$; TCEP: $n = 30$, $p = 1 \times 10^{-11}$; HA: $n = 15$, $p = 9.4 \times 10^{-10}$). Data are presented as box plots with the center line at the median, lower bound at 25th percentile, upper bound at 75th percentile, and whiskers at minimum and maximum values. (C) Schematic representation of acyl-biotin exchange (ABE) for detecting GSDMD palmitoylation. After treatment with NEM to block free thiols, the protein was incubated with HA to break acylated thiols and generate free thiols. Free thiols were then biotinylated by thiol-reactive biotin and biotinylated protein was pulled-down by streptavidin beads. Specific proteins were detected by immunoblot. (D) Palmitoylation of recombinant full-length mGSDMD (molecular weight of 55 kDa) detected by ABE. The assay was done in the presence of Yeast lysate and palmitoylation of the protein Vac8 (60 kDa) was verified as a positive control. Only regions of the blot with bands of interest are shown for clarity. Statistics were measured by Student's t-tests with **** for $p < 0.0001$. Source data are available online for this figure.

suggesting that palmitoylation is required for GSDMD binding to membranes.

Thus, our data show that palmitoylation is an essential prerequisite for the targeting of GSDMD to the membrane. However, it is known that negatively charged lipids, and particularly PIPs, play a crucial role in GSDMD activity (Ding et al, 2016; Liu et al, 2016; Mulvihill et al, 2018; Sborgi et al, 2016), supposedly by favoring the initial binding step via interaction with GSDMD basic patches (Xia et al, 2021). Therefore, we asked if palmitoylated GSDMD yet requires the contribution of specific lipids to bind and assemble oligomers at the membrane. To answer this question, we performed SMI of palmitoylated GSDMD on SLBs made of neutral Egg PC, without the negatively charged lipid PI(4,5)P$_2$, which we normally include in our SLB lipid composition (Fig. EV3). According to our experimental protocol, we added

active mGSDMD-mEGFP to the preformed SLBs and allowed them to equilibrate for one hour, followed by washing to remove unbound protein. Compared to samples containing PI(4,5)P$_2$, Egg PC SLBs had a significantly lower density of GSDMD particles (Fig. EV3A,B), indicating that PI(4,5)P$_2$ contributes to GSDMD membrane binding. Of note, the small fraction of GSDMD particles associated with the Egg PC membrane was mainly immobile, suggesting membrane insertion (Fig. EV3C). These particles exhibited a narrow intensity distribution, resulting prevalently in the presence of monomers and dimers (Fig. EV3D,E). Overall, these results point to a role of PI(4,5)P$_2$ in aiding GSDMD initial membrane binding and supposedly oligomerization. They also show that palmitoylation is necessary but not sufficient to stably bind GSDMD to the membrane and PI(4,5)P$_2$ plays a synergistic role.

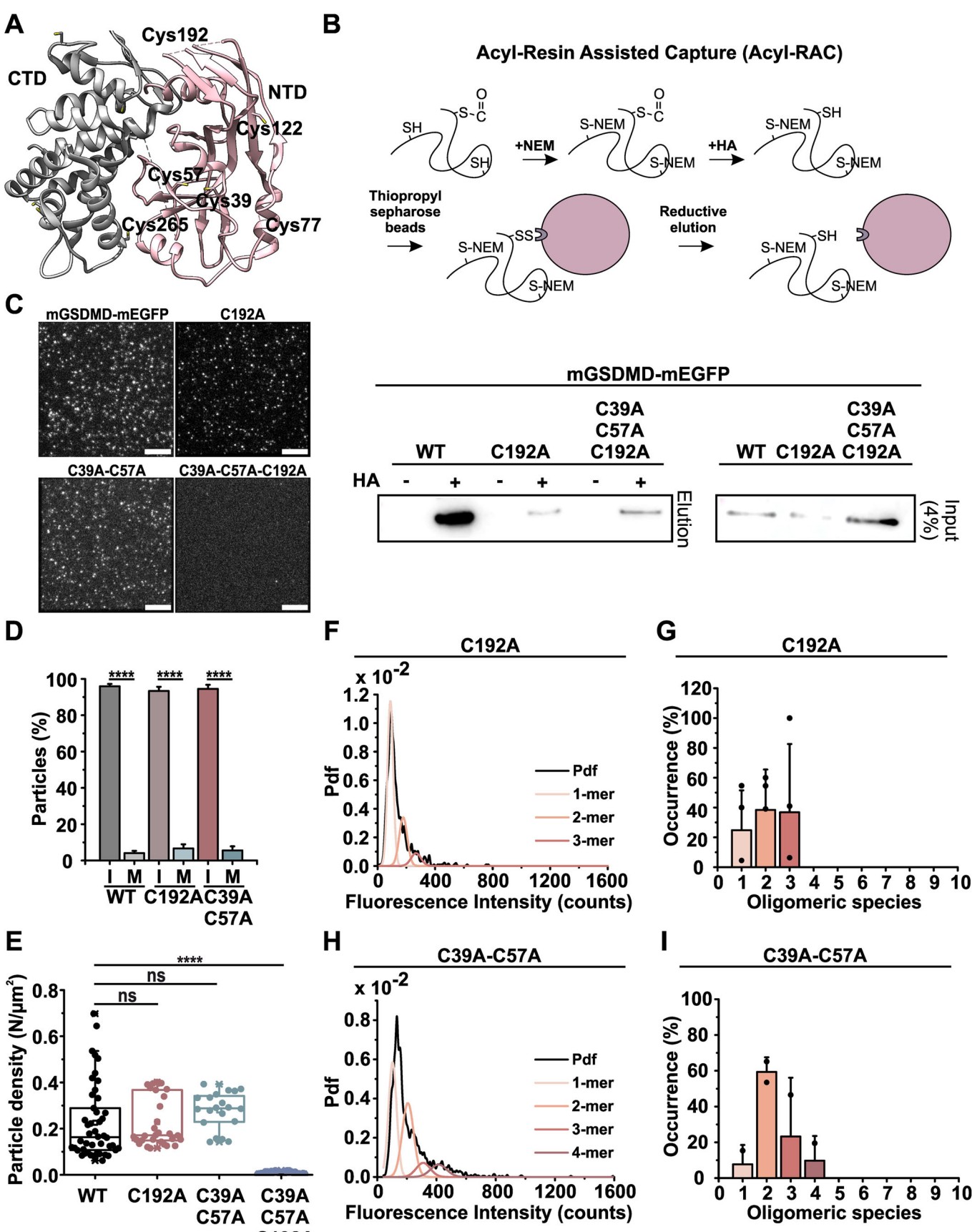

◄ **Figure 4.  Cysteines control palmitoylation-mediated membrane binding and oligomerization.**

(A) Structure of mGSDMD FL (modeled based on PDB 6N9N (Liu et al, 2019)). Cysteine residues are presented as sticks and mutated residues on the NTD are highlighted in black. (B) Schematic representation of Acyl-Resin Assisted Capture (Acyl-RAC) for detecting GSDMD palmitoylation. After treatment with NEM to block free thiols, the protein was incubated with HA to break acylated thiols and generate free thiols. The protein was then incubated with thiopropyl sepharose beads and pulled-down by reductive elution (up). mGSDMD-mEGFP palmitoylation was detected by immunoblot (bottom) comparing WT, C192A, and the triple mutant. Only regions of the blot with bands of interest are shown for clarity. (C) Representative TIRF images of preformed SLB incubated with 2 nM active mGSDMD-mEGFP or cysteine mutants (C192A, C39A-C57A, and C39A-C57A-C192A) for 1 h. Scale bars 5 μm. (D) Percentage of immobile (I) versus mobile (M) particles detected in the mGSDMD-mEGFP WT, mGSDMD-C192A-mEGFP, and mGSDMD-C39A-C57A-mEGFP samples. Averages from at least two independent experiments with a minimum of 10 recordings analyzed for each experiment (WT: $p < 0.0001$; C192A: $p < 0.0001$; C39A-C57A: $p < 0.0001$). (E) Density of detected assemblies on SLB comparing active mGSDMD-mEGFP with the C192A mutant and the C39A-C57A double mutant as in (C). Plotted are values obtained from a minimum of 10 recordings per experiment, from at least two independent experiments (C192A: $p = 0.93$; C39A-C57A: $p = 0.17$; C39A-C57A-C192A: $p = 5.4 \times 10^{-9}$). Data are presented as box plots with the center line at the median, lower bound at 25th percentile, upper bound at 75th percentile, and whiskers at minimum and maximum values. (F) Representative fluorescence distributions of mGSDMD-C192A-mEGFP oligomers obtained from preformed SLB incubated with 2 nM active protein. The resulting brightness distribution was plotted as a probability distribution function (Pdf, black) and fitted with a mixture of Gaussians to estimate the percentage of occurrence oligomers (color). (G) Percentage of occurrence of mGSDMD-C192A-mEGFP oligomeric species calculated as the average value from four experiments with a minimum of 2500 particles per experiment (the color code used for the occurrence graphs is the same as for the distribution in (F)). Data are corrected for GFP partial labeling. Individual experimental data points are indicated as scatter plots in the graphs (0 values are not indicated). (H) Representative fluorescence distributions of mGSDMD-C39A-C57A-mEGFP oligomers obtained from preformed SLB incubated with 2 nM active protein. The resulting brightness distribution was plotted as a probability distribution function (Pdf, black) and fitted with a mixture of Gaussians to estimate the percentage of occurrence oligomers (color). (I) Percentage of occurrence of mGSDMD-C39A-C57A-mEGFP oligomeric species calculated as the average value from two experiments with a minimum of 3000 particles per experiment the color code used for the occurrence graphs is the same as for the distributions in (H)). Data are corrected for GFP partial labeling. Individual experimental data points are indicated as scatter plots in the graphs (0 values are not indicated). Error bars represent SD from the different experiments. Statistics were measured by Student's t-tests with **** for $p < 0.0001$ and ns (non-significant) for $p > 0.05$. Source data are available online for this figure.

## Cys192 is palmitoylated and controls GSDMD membrane insertion together with Cys39 and Cys57

Next, we wanted to find out which cysteines are involved in palmitoylation-mediated membrane targeting. Mouse GSDMD contains six cysteines in its NTD (Fig. 4A). Of these, Cys192 (Cys191 in human) has been identified as key for GSDMD oligomerization and mediated cell death (Devant et al, 2023; Liu et al, 2016; Rathkey et al, 2018). We therefore hypothesized that this cysteine might be responsible for membrane binding. To test this hypothesis, we produced recombinant mGSDMD-mEGFP with a C192A mutation (mGSDMD-C192A-mEGFP). Liposome leakage assay confirmed that this mutation impairs GSDMD permeabilization activity (Fig. EV4A). Presumably, this could be due to the inability of this mutant to bind the membrane as it lacks the cysteine responsible for GSDMD palmitoylation. We then performed an Acyl-RAC assay on mGSDMD-C192A-mEGFP. Acyl-RAC involves the sequential treatment of the protein with NEM and HA, as for the ABE assay, but it utilizes thiopropyl sepharose beads and reductive elution for the protein pull-down (Fig. 4B). The C192A mutant revealed a drastic reduction in palmitoylation pointing out that Cys192 is the most palmitoylated cysteine in GSDMD. Four studies published while the current work was in revision, confirm our findings (Balasubramanian et al, 2024; Du et al, 2024; Liu et al, 2024; Zhang et al, 2024).

Surprisingly, however, our SMI analysis revealed the presence of immobile fluorescent particles after the addition of activated mGSDMD-C192A-mEGFP on preformed SLBs, indicating protein binding and insertion into the membrane (Fig. 4C,D). Indeed, the number of membrane-inserted particles was not significantly different from that of mGSDMD-mEGFP WT (Fig. 4E). Noticing a faint residual palmitoylation band in the C192A mutant (Fig. 4B), we then surmised that other cysteines could mediate membrane association via palmitoylation. In a further attempt to identify them, we mutated and characterized all the other cysteines (Cys39, Cys57, Cys77, Cys122, and Cys265) present in the NTD of mGSDMD. Liposome leakage assay showed that C77A, C122A, and C265A mutants did not impair protein activity (Appendix Fig. S3).

C39A and C57A mutants reduced GSDMD-mediated membrane permeabilization, although to a minor extent compared to C192A and with a comparable particle density to the WT (Fig. EV4A–D). Alike Cys192, Cys39, and Cys57 are conserved between human and mouse. Cys39 plays a role in GSDMD oligomerization, although minor compared to Cys192 (Liu et al, 2016; Rathkey et al, 2018), while Cys57 is a known easy target for several PTMs (Devant et al, 2023; Humphries et al, 2020; Wang et al, 2019). Interestingly, the C39A-C57A double mutant showed similar behavior to C192A in GSDMD-mediated liposome permeabilization and insertion into the membrane (Figs. EV4E and 4C,D), with a comparable particle density to the WT (Fig. 4E). Intriguingly, however, only simultaneous mutation of all three conserved cysteines (mGSDMD-C39A-C57A-C192A-mEGFP) abolished liposome permeabilization (Fig. EV4F) and resulted in complete impairment of protein insertion into the SLB after incubation of the activated protein (Fig. 4C,E). We then wondered if the minimal palmitoylation detected in the single C192A mutant was to attribute to Cys39 and Cys57. However, Acyl-RAC of the triple mutant still showed the same residual band indicating that this is not biologically relevant to membrane interaction (Fig. 4B). Hence, our data indicate that Cys192 through its palmitoylation, and Cys39 and Cys57 through a yet undetermined mechanism, contribute to GSDMD membrane insertion, as only the triple mutant is fully impaired in this process. Importantly, Cys39, Cys57, and Cys192 are the only cysteines required for GSDMD pore activity.

## Cys192 is sufficient for dimer/trimer formation but it requires Cys39 and Cys57 for higher-order oligomerization

Surprisingly, the reduced GSDMD permeabilization activity of the C192A mutant (Fig. EV4A) did not correlate with impaired GSDMD membrane insertion, as the particle density at the membrane was comparable to that of the WT (Fig. 4E). However, we observed that the particles of this mutant appeared less bright than in the WT and that their fluorescence intensity distribution showed a narrower curve (Fig. 4F and cf. Fig. 2E). Intriguingly,

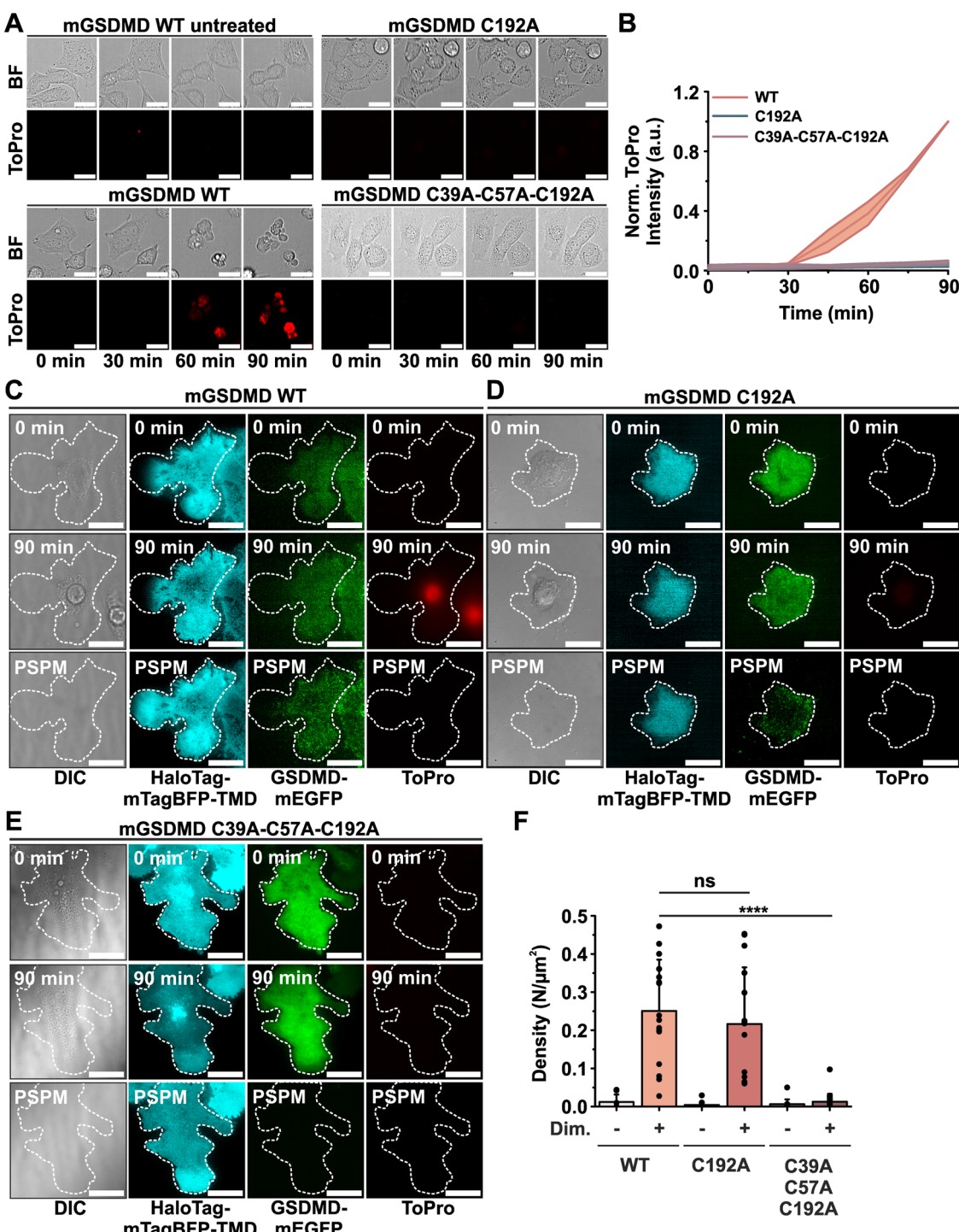

stoichiometry analysis revealed striking differences compared to the WT, as mGSDMD-C192A-mEGFP was unable to assemble into high-order oligomers, but predominantly formed dimers and trimers with a lower proportion of monomers (Fig. 4G). Overall, these data indicate that Cys192 is crucial for GSDMD high-order oligomerization, but not for initial assembly into small oligomers. Unlike the C192A, the single C39A and C57A mutations could not prevent the formation of high-order oligomers, although they could

reduce their production (Appendix Fig. S4A,B; Fig. EV4G,H). However, the combined C39A-C57A double mutant, showed a narrow intensity distribution similar to the C192A, with slightly higher values (Fig. 4H). This translated into the presence prevalently of dimers and trimers, with a smaller percentage of monomers and tetramers, but not higher-order oligomers (Fig. 4I). The mutants C77A, C122A, and C265A showed density and oligomerization comparable to the WT (Appendix Fig. S4C–I).

**Figure 5.  Mutation at multiple cysteines abrogates GSDMD membrane targeting in cells.**

(A) Representative confocal microscopy images of morphological changes (Bright Field - BF) of HEK293T cells stably transfected with mCaspase 1 and transfected with mGSDMD-mEGFP (WT, C192A and C39A-C57A-C192A), treated with 0.5 μM ToPro3-Iodide (ToPro, red) before and 30, 60 or 90 min after pyroptosis induction. Scale bars 20 μm. (B) Quantification of PM permeabilization of HEK293T cells shown in (A) by normalized fluorescence intensity of ToPro3-Iodide over 90 min ($n = 3$ experiments with >25 cells analyzed per experiment and condition). Lines in the graph correspond to the average values from all measured cells and colored areas to data variability (mean ± SD). (C–E) Representative TIRF images of HEK293T cells stably transfected with mCaspase 1 and transfected with mGSDMD-mEGFP WT (C), mGSDMD-mEGFP C192A (D), and C39A-C57A-C192A (E) (green), and HaloTag-mTagBFP-TMD (cyan) for stable tethering on a PLL-PEG-HTL-coated surface. Images after pyroptosis induction (90 min) indicated by ToPro3-Iodide oligomers (red), morphological changes (DIC), and mGSDMD-mEGFP oligomers (green), and after PSPM formation. Scale bars 20 μm. (F) Density of detected mGSDMD-mEGFP clusters in PSPMs as shown in (C–E) calculated as the average value from at least two independent experiments (WT + Dimerizer: 14 cells, 3 experiments; WT − Dimerizer: 8 cells, 2 experiments; C192A + Dimerizer: 13 cells, 3 experiments; C192A − Dimerizer: 16 cells, 3 experiments; C39A-C57A-C192A + Dimerizer: 22 cells, 3 experiments; C39A-C57A-C192A − Dimerizer: 14 cells, 3 experiments; C192A: $p = 0.51$; C39A-C57A-C192A: $p = 9.75 \times 10^{-10}$). Individual experimental data points are indicated as scatter plots in the graphs. Error bars represent SD from the different experiments. Statistics were measured by Student's t-tests with **** for $p < 0.0001$ and ns (non-significant) for $p > 0.05$. Source data are available online for this figure.

These results clearly show that both Cys192 and the Cys39/Cys57 pair are required for the formation of functional higher-order oligomers of GSDMD. Remarkably, both inactive GSDMD mutants, C192A and C39A-C57A, were still able to assemble into dimers and trimers, suggesting that dimers and trimers are not sufficient to cause membrane permeabilization, which requires assembly into larger protein complexes.

## Mutation at multiple cysteines abrogates GSDMD membrane targeting in cells

To validate the role of these cysteines in the context of the cell, we transfected HEK293T cells (lacking endogenous GSDMD and carrying a DmrB– mouse Caspase 1 construct under the control of a Dox-inducible promoter for pyroptosis induction (Rühl et al, 2018)) with mGSDMD-mEGFP variants harboring either C192A or C39A-C57A-C192A mutations and compared them with HEK293T cells stably expressing WT mGSDMD-mEGFP. The expression levels of the transiently transfected mutants were higher than those of the stably expressed WT (Fig. EV5A). However, in contrast to WT GSDMD, the cells expressing the mutant variants remained viable after pyroptosis induction, as examined by morphological analysis and plasma membrane permeabilization indicated by uptake of a PM-impermeable dye using confocal microscopy (Fig. 5A,B; Appendix Fig. S5A). As these mutants retained the ability to be cleaved by Caspase 1 (Appendix Fig. S5B), these results indicate that these cysteines are critical for GSDMD pore formation and PM permeabilization.

To better assess the membrane association capabilities of the different mutants, we used an approach developed by us to visualize GSDMD oligomers on the PM of pyroptotic cells by TIRF microscopy (preprint: Kappelhoff et al, 2023). Specifically, we immobilized cells undergoing pyroptosis on a polymer support (PS) and subsequently removed the cell body to obtain an intact native PM (called PSPM), incorporating GSDMD oligomers. This approach made it possible to remove the entire fluorescence contribution of fluorescently labeled cytosolic GSDMD, which would otherwise hinder the clear visualization of individual GSDMD oligomers on the PM ((preprint: Kappelhoff et al, 2023) and Fig. 5C–E). Accordingly, WT mGSDMD-mEGFP-expressing cells undergoing pyroptosis showed a heterogeneous GFP signal at the PM as visualized by TIRF microscopy, indicating the presence of GSDMD clusters (Fig. 5C). However, only after removal of the cell body and production of PSPMs (visualized by a HaloTag-

mTagBFP-TMD construct used to bind the membrane to the cell support), we could clearly visualize individual GSDMD dots at the PM in dimerizer-treated, but not in untreated, samples (Fig. 5C; Appendix Fig. S5C). The mGSDMD-C192A-mEGFP inactive mutant showed a homogeneous GSDMD-GFP signal prior to cell body removal and PSPM formation, in both treated and untreated cells (Fig. 5D; Appendix Fig. S5D). Surprisingly, however, PSPMs from these cells displayed clear GSDMD dots only in the treated samples, indicating the ability of this mutant to associate to the PM upon pyroptosis induction (Fig. 5D; Appendix Fig. S5D), as quantified by particle density at the PM (Fig. 5F). Of note, these dots exhibited a weaker intensity compared to the WT (compare Fig. 5C and D), suggesting that GSDMD assembles into smaller oligomers. This explains the impaired visualization of these dots in the presence of the cytosolic fluorescent background prior to cell body removal (Fig. 5D). Importantly, we were also able to recapitulate this behavior in the human GSDMD variant, where mutation of Cys191 to Ala (hGSDMD-C191A-mEGFP) could not prevent association to the membrane, resulting in a similar particle density to the WT (Fig. EV5B,C; Appendix Fig. S5E,F). In contrast, pyroptosis induction in cells carrying either mGSDMD-mEGFP or hGSDMD-mEGFP with mutations of all three cysteines did not result in GSDMD-GFP signal at the PM, even after removal of the cell body and production of PSPMs, indicating complete impairment of membrane binding (Figs. 5E,F and EV5D,E; Appendix Fig. S5G,H).

## Discussion

Here, we have uncovered a novel cooperative function of multiple cysteines in mediating GSDMD membrane targeting and oligomerization, contributing to the fundamental understanding of GSDMD pore assembly mechanism. This was achieved using high-sensitivity SMI microscopy combined with stoichiometry analysis and PSPM technology, which enabled to visualize and quantify small GSDMD oligomers that are not resolvable with conventional fluorescence microscopy.

Our stoichiometry analysis shows that GSDMD preassembles into dimers and trimers at the membrane, and these oligomeric building blocks can either directly insert into the membrane or further oligomerize before insertion. Although we cannot rule out the possibility that, under different experimental conditions, GSDMD monomers or small preformed oligomers may also

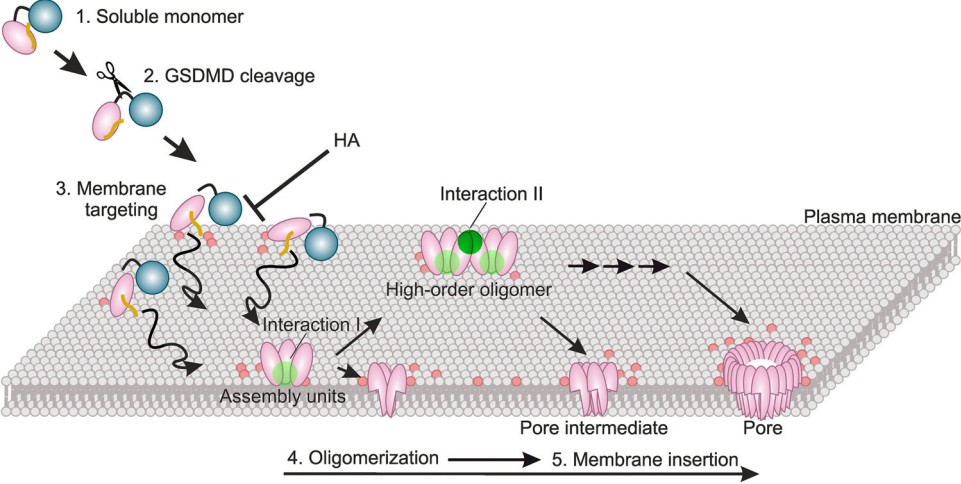

**Figure 6.    Model of GSDMD pore assembly at the membrane.**

GSDMD is present as monomers in solution (1). Cleavage by caspases (2) allows its targeting to the membrane mediated by Cys192 palmitoylation in synergy with negatively charged lipids (3). At the membrane, the CTD is released and the NTD assembles into dimeric and trimeric building blocks (assembly units) triggered by a first "transient interaction" (Interaction I) mediated by Cys39, Cys57, or Cys192. Dimers and trimers then either undergo the conformational changes required for the incorporation into the membrane or further oligomerize towards higher-order oligomers with a process that requires a second "cysteine-mediated transient interaction" (Interaction II), mediated by Cys39, Cys57, and Cys192 not involved in Interaction I. GSDMD-NTD is depicted in pink, GSDMD-CTD in blue, and palmitoylation in yellow. Negatively charged lipids at the PM are shown in red. "Cysteine-mediated interaction I" is represented as a light green dot and "cysteine-mediated interaction II" as a dark green dot.

assemble into higher-order oligomers within the membrane, our work proves that binding of GSDMD to the membrane, oligomerization, and final membrane insertion is a viable pathway for the assembly of GSDMD pores (Fig. 6). This model suggests that GSDMD assembly may occur prior to the conformational changes required for membrane insertion. This is consistent with the "prepore" conformation of GSDMD resolved by cryo-EM (Xia et al, 2021). However, GSDMD does not necessarily need to form a fully assembled ring to insert into the membrane, and intermediate oligomer structures are equally capable of breaching the membrane.

We found that GSDMD is extensively palmitoylated, mediating its targeting to membranes. How recombinant GSDMD is palmitoylated remains unclear since bacteria do not possess palmitoyltransferases. More likely, palmitoylation occurs via an autocatalytic mechanism, as shown for bacterial GSDMs (Johnson et al, 2022) and other proteins (Dietrich and Ungermann, 2004; Kümmel et al, 2010). Four studies published during the revision of this manuscript revealed a "regulated" ROS-dependent mechanism of GSDMD palmitoylation in myeloid cells, controlled by specific palmitoyltransferases, and an "unregulated" mechanism in recombinant and HEK cell-expressed GSDMD (Balasubramanian et al, 2024; Du et al, 2024; Liu et al, 2024; Zhang et al, 2024). We speculate that the regulated process might be more specific than the unregulated one, which can target cysteines more variably. Still, consistent with these studies, we identified Cys192 as the major palmitoylation target.

Another important finding of this study is that palmitoylation is necessary but not sufficient for GSDMD membrane association, which also requires the presence of PI(4,5)P$_2$. As Egg PC-only SLBs show a reduced number of palmitoylated GSDMD at the membrane, we propose that PIPs, and supposedly other negatively charged lipids, act synergistically with the palmitate group in

membrane association (Fig. 6). PIPs may facilitate both GSDMD targeting and oligomerization, as recent evidence suggests that these lipids may stabilize GSDMD pore opening (preprint: Kappelhoff et al, 2023; Santa Cruz Garcia et al, 2022).

Through comprehensive stoichiometry analysis of cysteine-to-alanine mutations, we identified Cys39, Cys57, and Cys192 as key cysteines in GSDMD pore assembly. Only the combined C39A-C57A-C192A mutations, but not single C192A or C39A-C57A double mutations, completely impaired the insertion of GSDMD into lipid bilayers. Importantly, we recapitulated this effect in the physiological PM environment for both mouse and human GSDMD using our SMI approach with PSPM technology, which visualizes GSDMD oligomers at the PM with high sensitivity. This finding is remarkable as it indicates a previously unexplored cooperative role of Cys39, Cys57, and Cys192 in GSDMD pore formation. As membrane insertion is the final step in GSDM pore assembly, we explain this effect as resulting from a cooperative action of membrane targeting and oligomerization mediated by these cysteines.

In particular, Cys192 alone (in the C39A-C57A mutant) is sufficient to form dimers/trimers, indicating that beyond palmitoylation, it plays an additional role in mediating oligomerization. This also reinforces the model of dimers/trimers as building blocks of GSDMD assembly. However, Cys192 alone is insufficient for higher-order oligomer formation and requires at least one additional cysteine (Cys39 or Cys57). When Cys39 or Cys57 are individually mutated, higher-order oligomers still form, but not as effectively as in the WT protein. This supports the contribution of Cys39 and Cys57 to the oligomerization process.

Cys39/Cys57 together have a role similar to Cys192 alone in GSDMD oligomerization. They can also form dimers/trimers in the absence of Cys192, but require Cys192 to form functional higher-

order oligomers, consistent with our recent structural findings in cells (preprint: Kappelhoff et al, 2023). As both C192A and C39A-C57A mutants exhibit impaired activity in liposome leakage assays, this provides a clear elucidation of how these cysteines control GSDMD pore formation.

These results also show that dimers and trimers can insert into the membrane but are insufficient to form functional pores for pyroptosis. Whether these small oligomers allow the passage of small molecules, such as ions, in the early phase of pyroptosis, as hypothesized elsewhere (de Vasconcelos et al, 2019; Schaefer and Hummer, 2022), remains to be clarified.

Together, our stoichiometry analysis suggests that Cys192 and Cys39/Cys57 have a redundant role in dimer/trimer formation, but they are all required for higher-order oligomer formation. Consistent with this, when Cys39, Cys57, and Cys192 are all mutated, the protein remains monomeric, unable to progress even to dimers or trimers, which are the minimal units capable of membrane insertion, resulting in no membrane-inserted GSDMD at all. Considering that cysteines do not form disulfide bonds in the crystal and cryo-EM structures (Liu et al, 2019; Xia et al, 2021), we propose that these three cysteines may participate in subtle transient thiol-mediated interactions and structural changes that facilitate oligomer formation. Supporting this, reducing agents and triple mutations prevent membrane insertion (Figs. 3A and 4C).

Based on our findings, we propose a model for how Cys192-palmitoylation and Cys39, Cys57, and Cys192-mediated oligomerization cooperate in GSDMD pore assembly (Fig. 6). Initially, Cys192 palmitoylation promotes stable GSDMD membrane anchoring (synergistically with lipids), increasing local concentration and favoring a protein conformation conducive to oligomerization. Without palmitoylation (mutation of Cys192), the loss of anchoring weakens GSDMD membrane interaction and reduces its local concentration, preventing oligomer formation beyond dimers and trimers. The oligomerization process, involving all three cysteines, relies on a two-step "cysteine-mediated transient interaction" mechanism: Interaction I (mediated either by Cys192 alone or Cys39/57) accounts for the formation of dimers and trimers as assembly units, and Interaction II (possibly mediated by the cysteines not involved in Interaction I) allows oligomerization to proceed to higher-order oligomers. Of note, membrane insertion follows oligomerization and can occur at any time after dimer/trimer formation.

Despite Cys39 and Cys57 in the GSDMD-C192A mutant should be sufficient to satisfy our two-step model, the reduced GSDMD membrane presence (due to lack of palmitoylation) likely limits its oligomerization. This suggests that Cys192 has a more potent function compared to Cys39 and Cys57 alone, as it additionally contributes to oligomerization via its palmitoylation, explaining its dominant role in driving GSDMD-mediated pyroptosis (Hu et al, 2020a; Jiang et al, 2024; Rathkey et al, 2018).

Overall, our results reveal a sophisticated cysteine-mediated mechanism for GSDMD assembly within the membrane, extending beyond Cys192 palmitoylation and involving additionally Cys39 and Cys57. This work provides a definitive clarification of the role of cysteines in GSDMD pore formation, identifying them as key mediators of GSDMD membrane targeting and oligomerization. Importantly, we demonstrate a correlated role of cysteines, suggesting that blocking Cys192, as done by all current GSDMD

inhibitors, may not be sufficient to completely abrogate GSDMD activity at membranes and control pyroptosis in disease.

# Methods

### Reagents and tools table

| Reagent/Resource | Reference or Source | Identifier or Catalog Number |
|---|---|---|
| **Experimental Models** | | |
| DmrB-mCasp1-transgenic HEK293T cells (*H. sapiens*) | Gift from P. Broz, Basel (Rühl et al, 2018) | N/A |
| DmrB-mCasp1 + mGSDMD-mEGFP-transgenic HEK293T cells (*H. sapiens*) | This study | N/A |
| *Electrocompetent Escherichia coli* BL21 RIPL | Gift from A. García-Sáez, Cologne | N/A |
| **Recombinant DNA** | | |
| pET21a-mGSDMD-8xHis | This study | N/A |
| pET21a-mGSDMD-mEGFP-8xHis | This study | N/A |
| pET21a-mGSDMD-C39A-mEGFP-8xHis | This study | N/A |
| pET21a-mGSDMD-C57A-mEGFP-8xHis | This study | N/A |
| pET21a-mGSDMD-C77A-mEGFP-8xHis | This study | N/A |
| pET21a-mGSDMD-C122A-mEGFP-8xHis | This study | N/A |
| pET21a-mGSDMD-C192A-mEGFP-8xHis | This study | N/A |
| pET21a-mGSDMD-C265A-mEGFP-8xHis | This study | N/A |
| pET21a-mGSDMD-C39A-C57A-mEGFP-8xHis | This study | N/A |
| pET21a-mGSDMD-C39A-C57A-C192A-mEGFP-8xHis | This study | N/A |
| pSems-mGSDMD-mEGFP | This study | N/A |
| pSems-mGSDMD-C192A-mEGFP | This study | N/A |
| pSems-mGSDMD-C39A-C57A-C192A-mEGFP | This study | N/A |
| pSems-hGSDMD-mEGFP | This study | N/A |
| pSems-hGSDMD-C191A-mEGFP | This study | N/A |
| pSems-hGSDMD-C38A-C56A-C191A-mEGFP | This study | N/A |
| **Antibodies** | | |
| Rabbit anti-GSDMD | Abbexa | Cat#Abx136074 |
| Rabbit anti-Vac8 | Ungermann laboratory – Osnabrück University | N/A |
| Mouse anti-β-Actin | Sigma | Cat#A5316 |
| Goat anti-rabbit HRP | Jackson ImmunoResearch | Cat#111-035-003 |

| Reagent/Resource | Reference or Source | Identifier or Catalog Number |
|---|---|---|
| Goat anti-mouse HRP | Jackson ImmunoResearch | Cat#115-035-003 |
| **Oligonucleotides and other sequence-based reagents** | | |
| PCR primers | This study | Appendix Table S1 |
| **Chemicals, Enzymes and other reagents** | | |
| dNTP Mix | NEB | Cat#N0047S |
| Q5 DNA High-Fidelity Polymerase | NEB | Cat#M0491L |
| T4 Polynucleotide Kinase | NEB | Cat#M0201S |
| T4 DNA Ligase | NEB | Cat#M0202S |
| Restriction endonucleases | NEB | N/A |
| MEM Eagle | PAN Biotech | Cat#P04-09500 |
| Tet-ON approved Fetal Bovine Serum (Tet-ON FBS) | Thermo Fisher Scientific | Cat#A4736201 |
| Non-essential amino acids (MEM NEAA) | PanBiotech | Cat#P08-32100 |
| HEPES buffer 1M | PanBiotech | Cat#P05-01100 |
| Trypsin/EDTA (10x) | PanBiotech | Cat#P10-024100 |
| DPS | PanBiotech | Cat#P04-35500 |
| Doxycycline | Sigma | Cat#D3447 |
| B/B-Homodimerizer | Takara Bio | Cat#AP20187 |
| ToPro3™-iodide | Thermo Fisher Scientific | Cat#T3605 |
| Latrunculin B | Abcam | Cat#Ab144291 |
| Protease inhibitor cocktail | SERVA | Cat#39106 |
| DNAseI | Sigma | Cat#DN25 |
| Lysozyme | Carl Roth | Cat#8259.2 |
| PMSF | Sigma | Cat#78830 |
| PureCube 100 INDIGO Ni-Agarose | Cube Biotech | Cat#75103 |
| Imidazole | Carl Roth | Cat#3899.4 |
| Nitrocellulose membrane | Cytiva Amersham™ Protan™ | Cat#10600006 |
| Powdered milk | Carl Roth | Cat#T145.2 |
| Phosphatidylcholine | Avanti Polar Lipids - Sigma | Cat#840051P |
| Phosphatidylinositol 4,5-bisphosphate | Avanti Polar Lipids - Sigma | Cat#840046P |
| Cardiolipin | Avanti Polar Lipids - Sigma | Cat#840012P |
| Polycarbonate membranes | Avanti Polar Lipids - Sigma | Cat#610005 |
| Dithiothreitol (DTT) | Carl Roth | Cat#6908.2 |
| Tris(2-carboxyethyl)phosphine hydrochloride (TCEP) | Sigma | Cat#C4706 |
| 2-mercaptoethanol (2ME) | Carl Roth | Cat#4227.3 |
| Hydroxylamine (HA) | Sigma | Cat#159417 |
| Hydrogen peroxide ($H_2O_2$) | Sigma | Cat#95294 |
| N-Ethylmaleimide (NEM) | Sigma | Cat#E3876 |

| Reagent/Resource | Reference or Source | Identifier or Catalog Number |
|---|---|---|
| EZ-Link Biotin-HPDP | Fisher Scientific | Cat#21341 |
| Thio-propyl sepharose beads | G-Biosciences | Cat#786-1785 |
| Calcein | Sigma | Cat#C0875 |
| Sephadex G50 | GE Healthcare | Cat#17-0043-01 |
| Ibidi® μ-slide 8-well | ibidi | Cat#80826 |
| Microscopy chips | Marienfeld Laboratory Glassware | Cat#0117640 |
| 96 well plates | Sarstedt | Cat#82.1581.120 |
| NucleoBond™ Xtra Midi | Macherey-Nagel™ | Cat#740410.100 |
| Protein Thermal Shift™ dye kit | Thermo Fisher Scientific | Cat#4461146 |
| **Software** | | |
| ImageJ/FIJI | https://doi.org/10.1038/nmeth.2019 | https://imagej.net/software/fiji/ |
| Acquire^MP & Discover^MP v2023 | Refeyn Ltd | N/A |
| Stoichiometry Analysis Software (SAS) | (Danial et al, 2022) | https://github.com/jdanial/SAS |
| SLIMfast 4C | (Bellón et al, 2022) | N/A |
| Picasso software suite | (Schnitzbauer et al, 2017) | https://github.com/jungmannlab/picasso |
| OriginPro 9.0 | OriginLab Corporation | https://www.originlab.com |
| **Other** | | |
| Plasma Cleaner femto 1A | Diener electronics | N/A |
| CellVoyager™ CQ1 Benchtop High-Content Analysis System | Yokogawa | N/A |
| cellTIRF-4-Line | Olympus | N/A |
| TIRF microscope | (Winkelmann et al, 2024) | N/A |
| Äkta go™ | Cytiva | N/A |
| Superdex 200 10/300 Increase | GE Healthcare | 28-9909-44 |
| Trans-blot SD semy-dry transfer cell | Biorad | N/A |
| AZURE 600 | Azure Biosystems, Biozym | N/A |
| Refeyn TwoMP | Refeyn Ltd | N/A |
| qTower³ | Analytik Jena | N/A |
| Mini-extruder | Avanti Polar Lipids | 610020 |
| Tecan Infinite 200 pro MPlex | Tecan | N/A |

## Methods and protocols

### Constructs

Full-length mouse GSDMD WT and mouse GSDMD-mEGFP were cloned into the pET21a vector with a C-terminal His8-tag and the C39A, C57A, C77A, C122A, C192A, and C265A single mutants, the

C39A-C57A double mutant and the C39A-C57A-C192A triple mutant were generated.

All plasmids were transformed into DH5α-competent cells and verified by sequencing after midiprep (NucleoBond™ Xtra Midi - 740410.100, Macherey-Nagel™). All mutations were introduced by PCR (Appendix Table S1).

### Mammalian cell culture and transfection

Human embryonic kidney 298T cells (HEK293T) stably transfected with DmrB-mCaspase 1 (Tet-On vector) and double-stable DmrB-mCaspase 1 (Tet-On) + mGSDMD-mEGFP were maintained at 37% and 5% $CO_2$ in MEM medium (PAN Biotech, P04-09500) containing 10% Tet-ON approved fetal bovine serum (Thermo-Fisher, A4736201), 1% non-essential amino acids (PanBiotech, P08-32100), and 1% HEPES buffer 1M (PanBiotech, P05-01100). Cells were transiently transfected with pSems-mGSDMD-C192A-mEGFP, pSems-mGSDMD-C39A-C57A-C192A-mEGFP, pSems-hGSDMD-mEGFP, pSems-hGSDMD-C191A-mEGFP, or pSems-hGSDMD-C38A-C56A-C191A-mEGFP at 80% confluency by calcium phosphate co-precipitation over-night. The day before microscopy, cells were detached by Trypsin/EDTA (PanBiotech, P10-024100) treatment and seeded on the specific microscopy support.

### Polymer-supported plasma membranes

Polymer-supported plasma membranes (PSPMs) were formed as previously described in (preprint: Kappelhoff et al, 2023). Briefly, microscopy chips (0117640, Marienfeld Laboratory Glassware) were cleaned with isopropanol before and after plasma cleaning (Plasma Cleaner femto 1A, Diener electronics) at 100% power for 15 min. Chips were coated with a mixture of poly-L-lysine coupled to a polyethylene glycol functionalized with HaloTag ligand (PPL-PEG-HTL) or RGD-peptide (PLL-PEG-RGD) in a ratio of 30/70% (w/w) respectively (Wedeking et al, 2015). Cells for PSPM generation were transfected with pDisplay-HaloTag-mTagBFP-TMD-GSlinker (TMD sequence: ASALAALAALAALAALAALAA-LAKSSRL) and seeded on the functionalized slides the day before microscopy. The RGD-peptide functionalization enables adherence of the cells by integrin interactions, while the covalent HTL-HaloTag interaction allows tethering of the PM to the surface. To obtain PSPMs, cells were then treated with 20 µM Latrunculin B (Abcam, ab144291) for 5 min at 37 °C and subsequently, the cell body was removed by sheer forces through heavy pipetting directly at the microscope. After washing the sample 3x with 1 ml PBS (DPBS, PanBiotech, P04-35500). PSPMs were fixed with 4% paraformaldehyde and imaged by TIRF microscopy.

### Pyroptosis induction

To induce expression of DmrB-mCaspase 1, around 16 h before the experiments, cells were treated with 500 ng/mL Doxycycline (D3447, Sigma). Pyroptosis was induced by adding 500 nM B/B-Homodimerizer (AP20187, Takara Bio) at 37 °C and 5% $CO_2$. To monitor plasma membrane permeabilization after pyroptosis induction, cells were treated with 0.5 µM ToPro™-3 Iodide (T3605, Thermo Fisher Scientific). PM permeabilization kinetics were performed by imaging cells before and every 15 min after pyroptosis induction at a Spinning disk confocal microscope (CellVoyager™ CQ1 Benchtop High-Content Analysis System,

Yokogawa) at 37 °C with a dry 40× objective in an 8-well slide (ibidi® µ-slide 8-well, 80826, Ibidi) coated with PLL-PEG-RGD.

### Cell lysates

Cells, untreated and after 90 min of pyroptosis induction, were collected and lysed using lysis buffer (50 mM Tris-HCl pH 7.4, 1% SDS) supplemented with protease inhibitor cocktail (SERVA, 39106). Cell lysates were incubated for 30 min on ice and then centrifuged at $5000 \times g$ for 15 min. The supernatant was then collected and subjected to immunoblot.

### Quantification of GSDMD oligomers in plasma membranes

Visualization of GSDMD oligomer formation in the plasma membrane of pyroptotic cells was performed at an inverted Olympus IX-81 microscope equipped with a motorized quad-line TIR-illumination condenser (cellTIRF-4-Line, Olympus), a motor-ized xy-stage (IM 120×80, Märzhäuser), a 100× oil immersion objective (UAPON 100x TIRF, NA 1.49, Olympus), a large incubator with temperature control (TempController 2000-2, CellVivo) and a $CO_2$-controller ($CO_2$-controller 2000, CellVivo). In order to ensure PM tethering by HaloTag anchor expression, mTagBFP excitation was achieved by a 405 nm diode laser (Olympus), mGSDMD-mEGFP oligomer formation was monitored by excitation with a 488 nm diode-pumped solid-state laser (Olympus), both set to TIR conditions. Cell permeabilization was monitored by nuclear staining by the cell-impermeable dye ToPro-3 Iodide via epi-mode excitation with a 640 nm diode laser (Olympus). Laser lines were filtered by clean-up filters (405 nm: BrightLine HC 390/40, Semrock, 488 nm: BrightLine HC 482/18, Semrock and 640 nm: BrightLine HC 640/1, Semrock). Fluorescence emission was filtered by a bandpass filter (BrightLine HC 446/523/500/677) and, in addition, single bandpass filters (Bright-Line HC 390/40, 482/18 and 640/14, Semrock) for each channel, respectively before detection with a sCMOS camera (ORCAFlash 4.0 V3, Hamamatsu). Images were acquired with the software CellSens 3.2 (Olympus) with an exposure time of 32 ms and $2 \times 2$ pixel binning, resulting in a pixel size of 130 nm. mGSDMD-mEGFP oligomers were detected using "Picasso Localize" (Schnitz-bauer et al, 2017). The Box size for the signal detection was set to 7 px and the Min. Net. Gradient. was adjusted according to the brightness of the signals in a range between 600 and 2500. Photon conversion parameters were set as follows: EM gain: 1; Baseline: 400; Sensitivity 0.46; Quantum efficiency 0.72, pixel size: 130 nm. The number of identified oligomers was normalized to the area of the cell to obtain the oligomer density.

### Protein purification

Plasmids encoding full-length mGSDMD WT, mGSDMD-mEGFP WT, and cysteine mutants were electroporated in *Escherichia coli* Codon-Plus, RIPL BL21 (DE3) cells and purified. Briefly, pellets from 4 L culture were lysed by sonication on ice in purification buffer (50 mM Tris-HCl pH 8, 150 mM NaCl, 1 mM DTT) + DNAseI (DN25, Sigma), lysozyme (8259.2, Carl Roth), PMSF (78830, Sigma), and protease inhibitors (SERVA, 39106). The 1 mM DTT had the only purpose to maintain protein stability and it did not interfere with the activity of the protein, as tested by calcein release assays. The clear lysate was incubated for 3 h with Ni-Agarose resin (PureCube 100 INDIGO Ni-Agarose – 75103,

Cube Biotech) at 4 °C under rotation. After incubation, the resin was sequentially washed with, in order, 20 column volume (CV) of purification buffer, 20 CV of purification buffer + 20 mM Imidazole (3899.4, Carl Roth), 20 CV purification buffer + 40 mM Imidazole and finally eluted with purification buffer + 500 mM Imidazole. After concentration, the elution was loaded on a Superdex 200 10/300 Increase size-exclusion column (28-9909-44, GE Healthcare) and purified by size exclusion chromatography (SEC) with an ÄKTA go™ system (Cytiva) in purification buffer.

Plasmid encoding deltaCARD Caspase 11 was transformed by heat shock in *Escherichia coli* BL21 RIPL cells and purified. Cell pellets were lysated by sonication on ice in caspase purification buffer (50 mM Tris-HCl pH 7.5, 150 mM NaCl, 20 mM imidazole) supplemented with 1 mM DTT, DNase, and protease inhibitors. The clear lysate was incubated with 1 mL Ni-Agarose resin. The resin was washed with 15 CV of lysis buffer and 15 CV of purification buffer containing 40 mM imidazole and the protein was eluted with 500 mM imidazole buffer. Eluted protein was collected and concentrated and SEC was carried out at 4 °C.

Fractions corresponding to the protein of interest were collected and protein quality was checked by SDS-polyacrylamide gel electrophoresis. All the purified proteins were aliquoted and snap-frozen in liquid nitrogen and stored at −80 °C.

All the mutants were tested for their quality and stability by thermal shift assay and SDS-PAGE showing no drastic changes in protein stability upon cysteine mutations (Appendix Fig. S6A–E). Moreover, this mutations did not affect caspase cleavage as proved by immunoblot (Appendix Fig. S6F).

### SDS-PAGE, NATIVE-PAGE, and immunoblot

For SDS-PAGE, protein samples were prepared in SDS sample buffer (150 mM Tris-HCl pH 7.5, 6% SDS, 0.2% Bromophenol Blue, 50% glycerol, and 1% β-mercaptoethanol), boiled at 95 °C for 10 min and loaded on 12% SDS gels and run with Tris-Glycine running buffer (192 mM glycine, 25 mM Tris, 0.1% SDS, pH 8.3). For non-reducing conditions the samples were prepared without reducing agents and the boiling step. For NATIVE-PAGE, protein samples were prepared in NATIVE sample buffer (100 mM Tris-HCl pH 6.8, 0.2% Bromophenol Blue, and 20% glycerol), loaded on 10% NATIVE gels, and run in NATIVE running buffer (192 mM glycine, 25 mM Tris, pH 8.3). For immunoblot, proteins were then transferred to a 0.2 μm nitrocellulose membrane (10600006 - Cytiva Amersham™ Protan™) at 0.2 A for 1 h in transfer buffer (192 mM glycine, 25 mM Tris, 10% methanol, pH 8.3) by using a trans-blot SD semy-dry transfer cell (Biorad). Blots were blocked with 5% milk (Carl Roth, T145.2) in TBST (Tris-buffered saline pH 7.6 + 0.1% Tween-20) and were probed with GSDMD primary antibody (1:500 - Rabbit polyclonal gasdermin-D antibody Abbexa, abx136074) in 5% milk blocking buffer overnight at 4 °C. For immunoblot controls, membranes were probed with Vac8 primary antibody (Ungermann laboratory - Osnabrück University) or mouse monoclonal β-actin primary antibody (1:1000 - Sigma, A5316). After incubation, blots were washed three times with 1x TBST and probed with a secondary HRP-conjugated antibody (1:2500 – Goat anti-rabbit IgG secondary antibody HRP – Jackson ImmunoResearch, 111-035-003 or 1:2500 – Goat anti-mouse IgG secondary antibody HRP – Jackson ImmunoResearch, 115-035-003) overnight at 4 °C.

Membranes were then washed with TBST and incubated for 1 min with ECL solution. The chemiluminescence signal was captured by an AZURE 600 gel imager (Azure Biosystems, Biozym).

### Mass photometry

Mass photometry measurements were performed using a Refeyn TwoMP mass photometer (Refeyn Ltd, Oxford, UK). Data acquisition and evaluation were performed using Acquire^MP and Discover^MP (both Refeyn Ltd, Oxford, UK), respectively.

Perforated silicone gaskets were placed on glass coverslips to form wells for every sample to be measured. Samples were acquired for 60 s at a final concentration of 50 nM in a total volume of 20 μl in the protein purification buffer used for SEC. Calibration was performed using β-amylase (Carl Roth, 8717.1).

### Protein thermal shift

To check the stability of all the purified proteins, the Protein Thermal Shift™ dye kit (4461146 - Thermo Fisher Scientific) was used. The method is based on the use of dyes to monitor thermal denaturation of proteins using a real-time PCR. Briefly, 20 μM of each protein was mixed with Protein Thermal Shift™ Dye and Protein Thermal Shift™ Buffer following manual instruction. A qTower^3 real-time thermal cycler (Analytik Jena) was used and a melting curve protocol was applied (Appendix Fig. S6A) starting at 25 °C, with an increasing temperature ramp up to 99 °C at 6 °C/s with $\Delta T = 1$ °C. The thermal denaturation profiles, melting temperatures, and the first derivative of the fluorescence emission as a function of temperature ($-dF/dT$) were calculated and plotted.

### Large unilamellar vesicles and supported lipid bilayers

All lipids were purchased from Avanti Polar Lipids. The desired lipid mixture, containing Egg phosphatidylcholine (Egg PC - 840051P) and Phosphatidylinositol 4,5-bisphosphate (PI(4,5)P₂ - 840046P) in a 99:1 molar (%) ratio, was dissolved in chloroform and the solvent was evaporated under nitrogen flux and vacuum dried for 3 h. The lipid mixture was then rehydrated to a final concentration of 0.7 mg/mL in SLB buffer (150 mM NaCl, 10 mM Hepes pH 7.4). To prepare large unilamellar vesicles (LUVs), the solution was passed through 10 cycles of freezing and thawing and manually extruded through a polycarbonate membrane of 100 nm pore size (Sigma, 610005) using glass syringes and the Avanti mini-extruder (Avanti Polar Lipids, Inc., Alabaster, AL). To obtain proteoliposomes, LUVs were incubated with purified mGSDMD-mEGFP at different concentrations: 2 nM vs 20 nM as indicated in the figures with the addition of 20 nM Caspase 11 at various time points from 2 min to 1 h. Lipid/protein ratios were ~$5 \times 10^{16}$ and $0.5 \times 10^{14}$ for 2 nM and 20 nM GSDMD, respectively.

After incubation, proteoliposomes were used to create supported lipid bilayers (SLBs) formed on glass slides freshly cleaned with Piranha solution. Briefly, glass microscopy chips were immersed into a 1:1 mixture of sulfuric acid (X9441, Carl Roth) with hydrogen peroxide (95294, Sigma) for at least 1 h. Slides were then rinsed with deionized water and sonicated in water for 5 min. Immediately before usage, chips were dried under a nitrogen stream. The SLB was formed by incubating proteoliposomes on a glass slide at 37 °C for 10 min with 3 mM CaCl₂, and washed 15 times with SLB buffer to remove non-fused vesicles. The 20 nM protein sample was diluted 1:10 with empty LUVs before SLB

formation to maintain single-molecule imaging. Alternatively, empty LUVs were incubated on piranha-cleaned chips at 37 °C for 10 min with 3 mM CaCl$_2$, and washed 15 times with SLB buffer. Then preformed SLBs were incubated with 0.5 nM, 1 nM or 2 nM pre-cleaved mGSDMD-mEGFP (WT or mutants) for 1 h and washed 15 times with SLB buffer. Lipid/protein ratio was $3 \times 10^{14}$ for the sample incubated with 2 nM GSDMD.

The treatment with hydrogen peroxide (H$_2$O$_2$ – 95294, Sigma), dithiothreitol (DTT – 6908.2, Carl Roth), Tris(2-carboxyethyl) phosphine hydrochloride (TCEP – C4706, Sigma), 2-mercaptoethanol (2ME – 4227.3, Carl Roth) and hydroxylamine (HA – 159417, Sigma) was performed by adding the different reagents for 30 min to the cleaved protein, before formation of the SLB.

### Single-molecule total internal reflection fluorescence (TIRF) microscopy

All samples were imaged on a custom-designed TIRF microscope (Winkelmann et al, 2024) for a total of 1500 frames under a 30 ms exposure time. Laser excitation from a 488 nm laser, ~200 W/cm$^2$ (Sapphire, Coherent) was coupled into a single mode polarization maintaining fiber to a TIRF module connected to an Olympus IX83 inverted microscope with hardware autofocus system (IX3-ZDC, Olympus) and 100× oil-immersion objective (UPLAPO100xOHR). The image was additionally magnified by 1.6× (IX3-CAS, Olympus) to obtain a final magnification of 160× and a pixel size of 100 nm. Fluorescence was filtered by a four-line polychroic mirror (zt405/488/561/640rpc, Chroma, 3 mm) and rejection band filter (zet405/488/561/647 TIRF, Chroma), and the emission was focused on an iXon Ultra EMCCD Camera (Andor Technologies).

### Stoichiometry analysis

We calibrated the fluorescence signal of the TIRF microscope before each experiment to avoid artifacts due to small changes in the optical setup. The calibration dataset necessary to calculate the monomeric intensity was obtained by sampling Halo-mEGFP incubated for 1 h on chips coated with a 99:1 mixture of poly-L-lysine coupled to a polyethylene glycol functionalized with either a CH$_3$OH group (PLL-PEG-OMe) or a Halo-tag ligand (PLL-PEG-HTL) respectively (Fig. EV1H). The oligomeric state of the protein in solution was analyzed with the same coated chips, treated with 2 nM Halo-GFP nanobody for 30 min. After treatment, chips were washed 15 times with SLB buffer and incubated with 1 nM mGSDMD-mEGFP for 1 h. After incubation with the protein, samples were washed 15 times with SLB buffer and imaged.

Images acquired by TIRF microscopy were analyzed by single-molecule brightness analysis using the Stoichiometry Analysis Software (SAS) (Danial et al, 2022). Briefly, bright spots were automatically detected using an implementation of the Difference of Gaussians method and thresholding. Selected particles were defined by a region of interest (ROI) of defined pixel size (pink circles in Fig. 1C) and fitted to two-dimensional (2D) Gaussians. Background subtraction was performed by defining a ROI around the particle's ROI having a larger pixel size. This algorithm provided the brightness value for each spot. Localized particles were filtered based on the distance between two ROIs, to avoid overlapping ROIs, and on the presence of multiple particles in the same ROI. The distribution of all obtained brightness values was plotted as a kernel probability density function (Pdf). The mean monomer intensity and standard deviation, calculated from monomers in the calibration step, were used to

determine the theoretical intensity of higher-order oligomeric species. This approach generated a linear combination of Gaussians, which was then used to fit the Pdf. Percentages of occurrence of GSDMD oligomeric species were calculated from the area of each fitted Gaussian. Values were corrected for partial labeling efficiency. Detection parameters were set as follows in SAS: camera pixel size (nm): 100, camera quantum efficiency: 95, camera offset: 170, camera EM gain (counts per photoelectron): 65.4, maximum sigma: 200. Analysis parameters were set as: labeling efficiency (for GFP): 70, maximum Gaussian mixtures: 30.

The number of identified oligomers for each recording, was normalized to the area of the acquisition field of view to obtain the particle density.

### Tracking analysis

Movies acquired were cut with Image J, and the first 100 frames were analyzed via tracking analysis by using the SLIMfast 4C software for localization-based imaging in Matlab (Bellón et al, 2022). Percentages of mobile and immobile particles were calculated and trajectories of mobile particles were examined (Movie EV1). Particles were defined as mobile if their displacement lasted for at least 5 frames while they were defined as immobile when they exhibited the same localization over time.

Control parameters were set as follows: camera pixel size (nm): 100, lag time (ms): 30, photon gain: 65.4, camera offset: 170, camera type: EMCCD, minimum immobile time (frame): 10, minimum mobile time (frame): 5, and evaluation cap size (frame): 0.

### Liposome leakage assay

The ability of mGSDMD to bind lipids and form pores in liposomes was determined by measuring the release of encapsulated self-quenching Calcein into LUVs. Briefly, Egg PC:Cardiolipin (CL - 840012P) 80:20 (molar %) LUVs were prepared as previously described but the lipid mixture was resuspended in 80 mM Calcein (C0875, Sigma) pH 7.0. After extrusion, LUVs were separated from free Calcein by Sephadex G50 beads (17-0043-01, GE Healthcare) in outside buffer (20 mM Hepes pH 7, 140 mM NaCl, 1 mM EDTA) in fractions of 250 μl. LUVs quality of each fraction was evaluated by fluorescence measurement of intact and permeabilized LUVs. Fluorescence (F) measurements were performed in black 96 well plates (82.1581.120, Sarstedt) in a final volume of 100 μL/well by using a Tecan plate reader Infinite 200 Pro M-Plex set at ex/em 490/520 wavelengths. LUV fractions with a permeabilized/intact fluorescence ratio of at least 5 were used in the experiment.

For the activity test, GSDMD WT and mutants at 62.5 nM (final concentration) and 10 nM Caspase 11 were mixed, at room temperature, to liposomes (50 mM total lipid) and immediately measured. For dose–response measurements, GSDMD from 0.1 nM to 150 nM (final concentrations) was incubated with 10 nM Caspase 11 for 15 min and then treated with 200 mM DTT or 10 μM H$_2$O$_2$ for 30 min before LUVs addition.

Calcein release was monitored every 2 min for 60 min. Afterward, each well was supplemented with 5 μl 5% Triton-100 in outside buffer to get the individual maximum fluorescence intensity for each well. LUV permeabilization was calculated as follows:

$$\% Calcein\ release = 100 \times \frac{[F_{Sample} - F_{only\ LUVs}]}{[F_{Triton} - F_{only\ LUVs}]}$$

### Detection of GSDMD palmitoylation by small-scale acyl-biotinyl exchange (ABE)

To detect GSDMD palmitoylation, the small-scale ABE assay from (Roth et al, 2006a) was performed. Briefly, 50–100 µg of purified protein was added to Yeast lysate up to 150 µL. Protein was collected by Chloroform/Methanol (CM) precipitation and denatured by dissolving the pellet in 30 µL solubilization buffer (SB - 4% SDS, 50 mM Tris-HCl, 5 mM EDTA pH 7.4) with 10 mM N-Ethylmaleimide (NEM - Sigma, E3876) for 10 min at 37 °C. Protein was diluted into 120 µL lysis buffer (LB - 150 mM NaCl, 50 mM Tris-HCl, 5 mM EDTA pH 7.4) containing 1 mM NEM and 0.2% Tergitol and incubated overnight with gentle mixing at 4 °C. The next day, three CM precipitations were performed to fully remove NEM. After the third precipitation, the resulting pellet was dissolved in 80 µL SB and split into two aliquots. The plus-hydroxylamine aliquot was diluted in 120 µL of 1 M hydroxylamine (HA - 159417, Sigma) pH 7.4, 150 mM NaCl, 0.2% Tergitol, and 1 mM EZ-Link Biotin-HPDP (21341, Fisher Scientific). The minus-hydroxylamine aliquot was prepared in the same way except that 50 mM Tris-HCl pH 7.4 substituted the HA. Samples were incubated for 1 h at 25 °C and protein was recovered by CM precipitation and dissolved in 40 µL TB (2% SDS, 50 mM Tris-HCl, 5 mM EDTA pH 7.4). 20 µL of the sample served as the input control while the other 20 µL was diluted into 800 µL of LB containing 0.2% Tergitol, and 0.1% SDS and incubated with 20 µL of High Capacity Streptavidin Agarose Resin (20357, Fisher Scientific) overnight at 4 °C. The next day, after four 1 mL washings with LB containing 0.2% Tergitol and 0.1% SDS, the protein was eluted in 50 µL elution buffer (5% SDS, 8 M urea, 40 mM Tris-HCl pH 6.8, 0.1 mM EDTA) and analyzed by SDS-PAGE and immunoblot.

### Detection of GSDMD palmitoylation by acyl-resin assisted capture (Acyl-RAC)

To compare palmitoylation of GSDMD WT versus cysteine mutants, the Acyl-RAC assay was performed. Briefly, 20 µg of purified protein was digested with Caspase 11 for 1 h on ice and mixed with BSA 10 mg/mL up to 150 µl. Protein was collected by Chloroform/Methanol precipitation and denatured by dissolving the pellet in 120 µl SB buffer + 10 mM NEM. Protein was then diluted in 480 µl of LB buffer + 10 mM NEM and 0.2% Tergitol and incubated overnight with gentle mixing at 4 °C. The next day, three CM precipitations were performed to fully remove NEM. After the third precipitation, the resulting pellet was dissolved in 400 µl SHB buffer (2% SDS, 5 mM EDTA, 100 mM HEPES pH 7.5) and diluted with 450 µl of Buffer A (5 mM EDTA, 100 mM HEPES pH 7.5). 30 µl of the mixture were collected as input sample. The rest of the volume was split into two equal parts of 400 µl and either 400 µl of 2 M HA (+HA) or 400 µl of 2 M Tris-HCl pH 8 (−HA) were added to the samples. To all the samples, 30 µl of Thio-propyl sepharose beads (786-1785 GBiosciences) were added and the mixture was incubated at 4 °C for 1 h with rotation. After three 1 mL washings with Buffer A + 1% SDS, protein was eluted by adding 40 µl of sample buffer + 1% β-mercaptoethanol and boiled for 10 min at 95 °C. Samples were then analyzed by Immunoblot.

## Data availability

Raw datasets and Source data are available at: https://doi.org/10.26249/FK2/48FTWW.

The source data of this paper are collected in the following database record: biostudies:S-SCDT-10_1038-S44318-024-00190-6.

## Peer review information

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

## Acknowledgements

We thank Petr Broz for providing HEK293T DmrB-mCaspase 1 cells and Christian Ungermann for providing Vac8 primary antibody. We also thank Changjiang You and Uris Ros for reading the manuscript and for helpful discussions. This work was supported by the German Research Foundation (DFG, SFB 944 and SFB 1557) to K.C., A.G.M., and R.K.

## Author contributions

**Eleonora Margheritis**: Conceptualization; Data curation; Formal analysis; Validation; Investigation; Visualization; Methodology; Writing—original draft; Writing—review and editing. **Shirin Kappelhoff**: Conceptualization; Formal analysis; Validation; Investigation; Visualization; Methodology; Writing—review and editing. **John Danial**: Software; Methodology; Writing—review and editing. **Nadine Gehle**: Investigation. **Wladislaw Kohl**: Investigation. **Rainer Kurre**: Resources; Data curation; Methodology; Writing—review and editing. **Ayelén González Montoro**: Conceptualization; Methodology; Writing—review and editing. **Katia Cosentino**: Conceptualization; Resources; Data curation; Supervision; Funding acquisition; Validation; Visualization; Methodology; Writing—original draft; Project administration; Writing—review and editing.

Source data underlying figure panels in this paper may have individual authorship assigned. Where available, figure panel/source data authorship is listed in the following database record: biostudies:S-SCDT-10_1038-S44318-024-00190-6.

## Funding

## Disclosure and competing interests statement

The authors declare no competing interests.

# Expanded View Figures

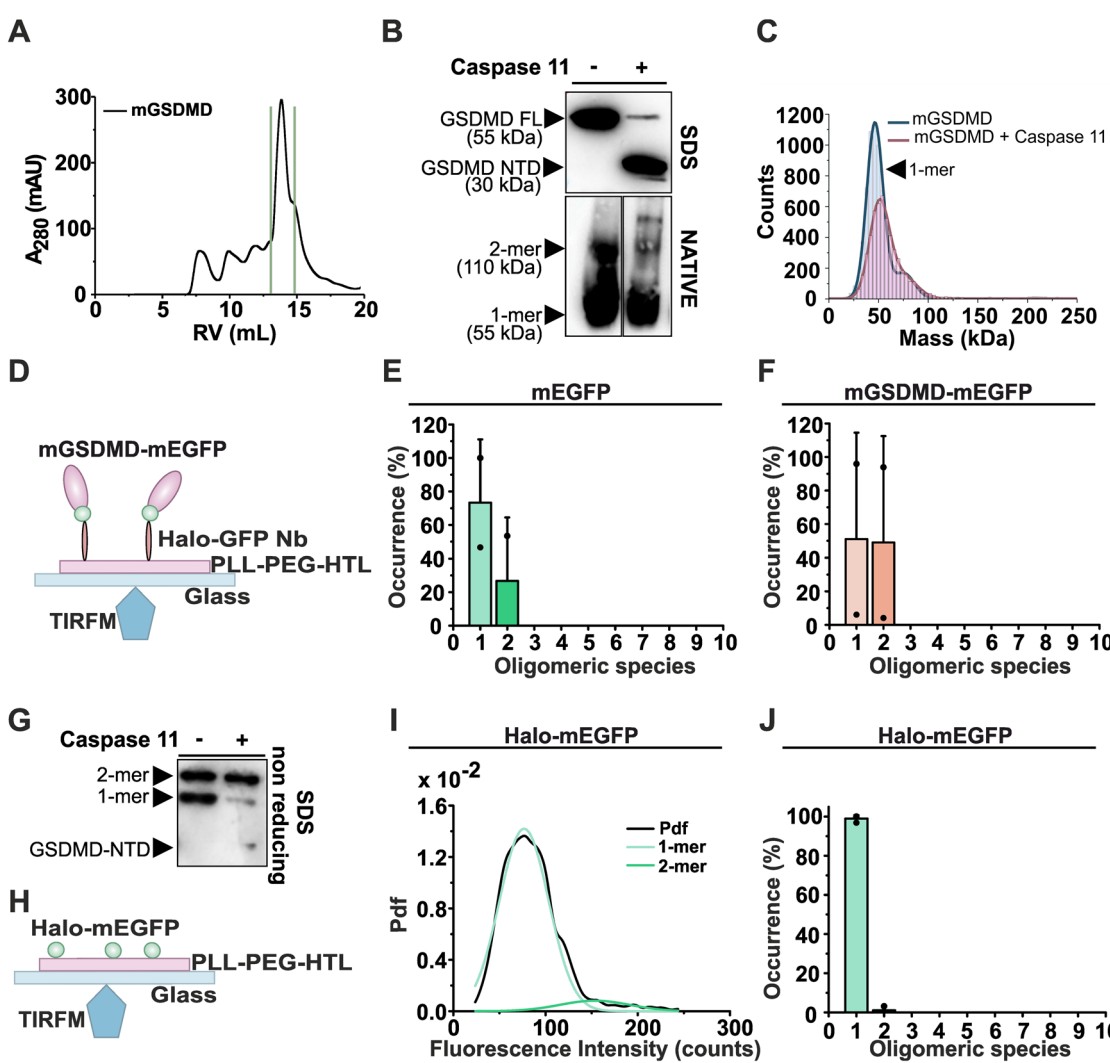

**Figure EV1. Purification of mouse GSDMD and characterization of the protein in solution.**

(A) Size-exclusion chromatography profile of recombinant full-length mouse GSDMD. (B) SDS and NATIVE immunoblots of full-length mouse GSDMD before and after incubation with Caspase 11. Only regions of the blot with bands of interest are shown for clarity. (C) Representative mass distribution of the oligomeric state of the protein before and after cleavage, analyzed by mass photometry. (D) Schematic representation of the sample for stoichiometry analysis of the protein in solution. The microscopy chip is functionalized with PLL-PEG-HTL (in pink) which allows the tethering of the Halo-GFP nanobody (in orange) used for the further capture of the mGSDMD-mEGFP protein (in pink and green) (see Methods). (E, F) Percentage of occurrence of mEGFP (E) and mGSDMD-mEGFP (F) oligomeric species in solution calculated as the average value from two different experiments with a minimum of 450 particles analyzed per experiment. Data are corrected for GFP partial labeling. (G) Immunoblot after SDS-PAGE in non-reducing conditions of full-length mouse GSDMD before and after incubation with Caspase 11. Only regions of the blot with bands of interest are shown for clarity. (H) Schematic representation of the system used to determine the fluorescence intensity of Halo-mEGFP used as calibration for the stoichiometric analysis. The microscopy chip is functionalized with PLL-PEG-HTL (in pink) which allows the tethering of Halo-mEGFP (green) (see Methods). (I) Representative fluorescence distribution of Halo-mEGFP obtained by incubating 10 pM Halo-mEGFP on a PLL-PEG-OMe: PLL-PEG-HTL 99:1 functionalized microscopy chip (see Methods). The resulting brightness distribution was plotted as a probability distribution function (Pdf, black) and fitted with a mixture of Gaussians to estimate the percentage of occurrence oligomers (color). (J) Percentage of occurrence of the oligomeric species of Halo-mEGFP detected confirming the monomeric state of the protein used as calibration for the stoichiometry analysis (the color code used for the occurrence graph is the same as for the distributions in (I)). Averages from three independent experiments with a minimum of 350 particles analyzed per experiment. Individual experimental data points are indicated as scatter plots in the graphs (0 values are not indicated). Error bars correspond to the SD from the different experiments.

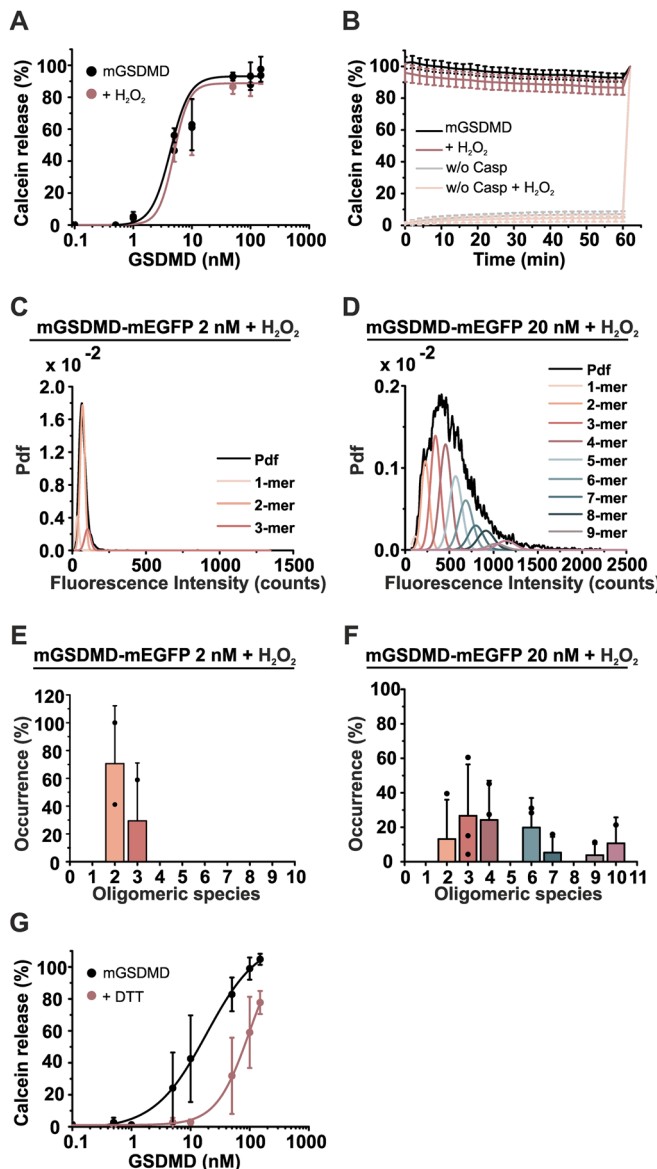

**Figure EV2. Oxidation does not affect the stoichiometry of GSDMD oligomers.**

(A) mGSDMD WT dose-dependent permeabilization of liposomes after 60 min incubation time, with or without 30 min incubation of the cleaved protein with 10 μM $H_2O_2$ before liposome addition. Represented are the averages from 3 independent experiments fitted with a Sigmoidal/Logistic fitting. (B) Liposome leakage assay as a percentage of calcein release for 60 min after incubation of liposomes with 62.5 nM mGSDMD WT with or without (w/o Casp) 10 nM Caspase 11 and pre-treated or not with 10 μM $H_2O_2$. Averages from three different experiments. (C, D) Representative fluorescence distribution of mGSDMD-mEGFP oligomers obtained from samples prepared with 2 nM (C) or 20 nM (D) active mGSDMD-mEGFP and incubated for 30 min with $H_2O_2$ before SLB formation. The resulting brightness distribution was plotted as a probability distribution function (Pdf, black) and fitted with a mixture of Gaussians to estimate the percentage of occurrence of particles containing n-mer oligomers (color). (E, F) Percentage of occurrence of the different oligomeric species of GSDMD detected (the color code used for the occurrence graph is the same as for the distributions in (E) and (F)). Averages from three independent experiments with a minimum of 2500 particles analyzed per experiment. Data are corrected for GFP partial labeling. Individual experimental data points are indicated as scatter plots in the graphs (0 values are not indicated). (G) mGSDMD WT dose-dependent permeabilization of liposomes after 60 min incubation time with or without 30 min incubation of the cleaved protein with 200 mM DTT before liposome addition. Averages from four independent experiments fitted with a Sigmoidal/ Logistic fitting. Error bars correspond to the SD from the different experiments.

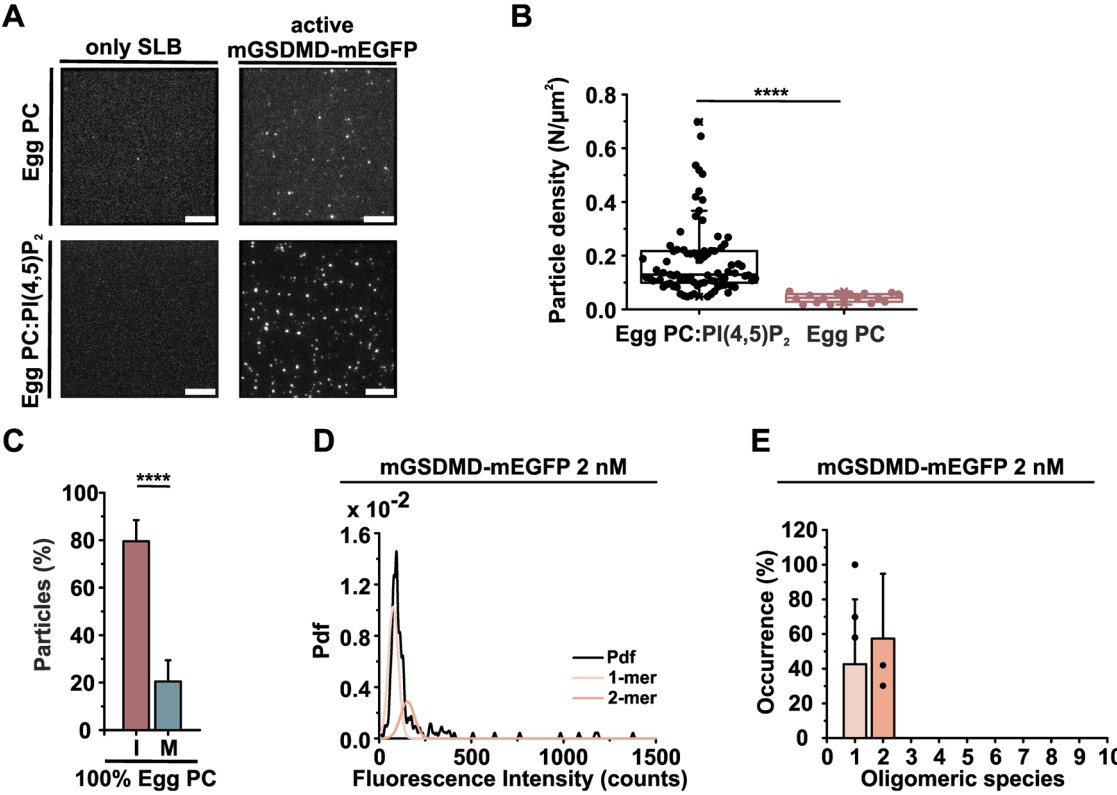

**Figure EV3. Negatively charged lipids contribute to membrane binding of palmitoylated GSDMD.**

(A) Representative TIRF images of SLB prepared with 100% Egg PC or Egg PC:PI(4, 5)P$_2$ 99:1 mol % (left) and incubated with cleaved mGSDMD-mEGFP (right) for 1 h. Scale bars 5 μm. (B) Density of detected particles on SLB prepared with 100% Egg PC or Egg PC:PI(4, 5)P$_2$ 99:1 mol %. Plotted are data from three different experiments (Egg PC:PI(4, 5)P$_2$ 99:1: $n = 85$; 100% Egg PC: $n = 20$; $p = 3.7 \times 10^{-5}$). Data are presented as box plots with the center line at the median, lower bound at 25th percentile, upper bound at 75th percentile, and whiskers at minimum and maximum values. (C) Percentage of immobile (I) versus mobile (M) particles detected in the 100% Egg PC. Averages from two independent experiments with a minimum of 10 recordings analyzed per experiment ($p = 1.03 \times 10^{-13}$). (D) Representative fluorescence distribution of mGSDMD-mEGFP oligomers obtained from preformed SLBs prepared with 100% Egg PC and incubated with 2 nM active protein. The resulting brightness distribution was plotted as a probability distribution function (black) and fitted with a mixture of Gaussians to estimate the percentage of occurrence of particles containing n-mer oligomers. (E) Percentage of occurrence of the different oligomeric species of GSDMD detected. Average from two independent experiments with a minimum of 300 particles analyzed per experiment (the color code used for the occurrence graph is the same as for the distributions in (D)). Data are corrected for GFP partial labeling. Individual experimental data points are indicated as scatter plots in the graphs (0 values are not indicated). Error bars correspond to the SD from the different experiments. Statistics were measured by Student's t-tests with **** for $p < 0.0001$.

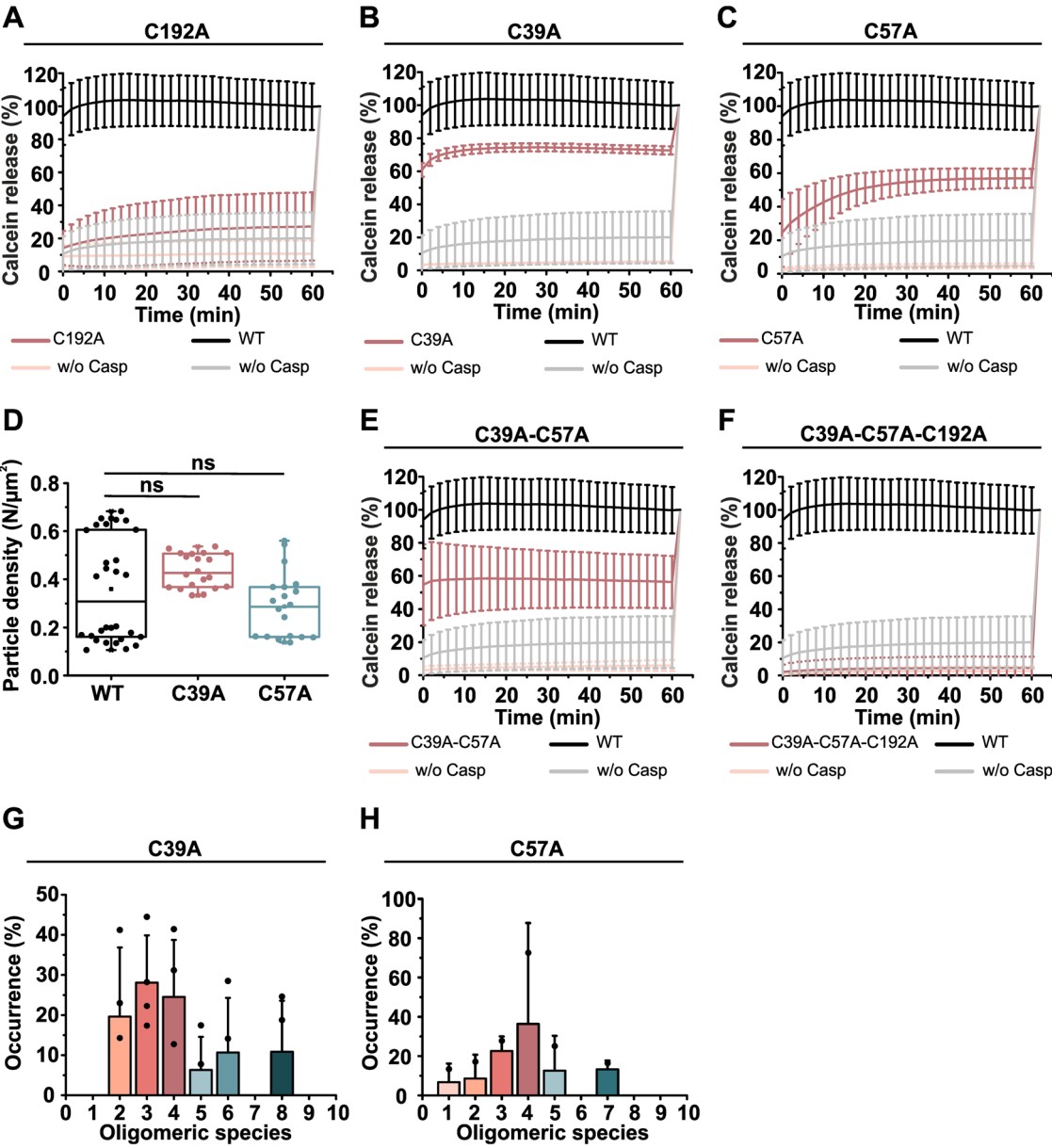

**Figure EV4. Permeabilization activity and stoichiometry analysis of cysteine mutants.**

(A–C) Liposome leakage assay as a percentage of calcein release for 60 min incubation with the cysteine mutants C192A (A), C39A (B), and C57A (C) with or without (w/o Casp) 10 nM Caspase 11 compared to the mGSDMD-mEGFP WT (WT) protein used as a reference for activity. Averages from at least two independent experiments. (D) Density of detected particles comparing active mGSDMD-mEGFP with the cysteine mutants C39A and C57A. Plotted are data from at least two different experiments (C39A: $p = 0.135$; C57A: $p = 0.196$). Data are presented as box plots with the center line at the median, lower bound at 25th percentile, upper bound at 75th percentile, and whiskers at minimum and maximum values. (E, F) Liposome leakage assay as a percentage of calcein release for 60 min incubation with the cysteine mutants C39A-C57A (E) and C39A-C57A-C192A (F) with or without (w/o Casp) 10 nM Caspase 11 compared to the mGSDMD-mEGFP WT (WT) protein used as a reference for activity. Averages from at least two independent experiments. (G, H) Percentage of occurrence of mGSDMD-C39A-mEGFP (G) and mGSDMD-C57A-mEGFP (H) oligomeric species calculated from at least two independent experiments with a minimum of 3000 particles per experiment (the color code used for the occurrence graph is the same as for the distributions in Appendix Fig. S4A,B). Data are corrected for GFP partial labeling. Individual experimental data points are indicated as scatter plots in the graphs (0 values are not indicated). Error bars correspond to the SD from the different experiments. Statistics were measured by Student's t-tests with ns (non-significant) for $p > 0.05$.

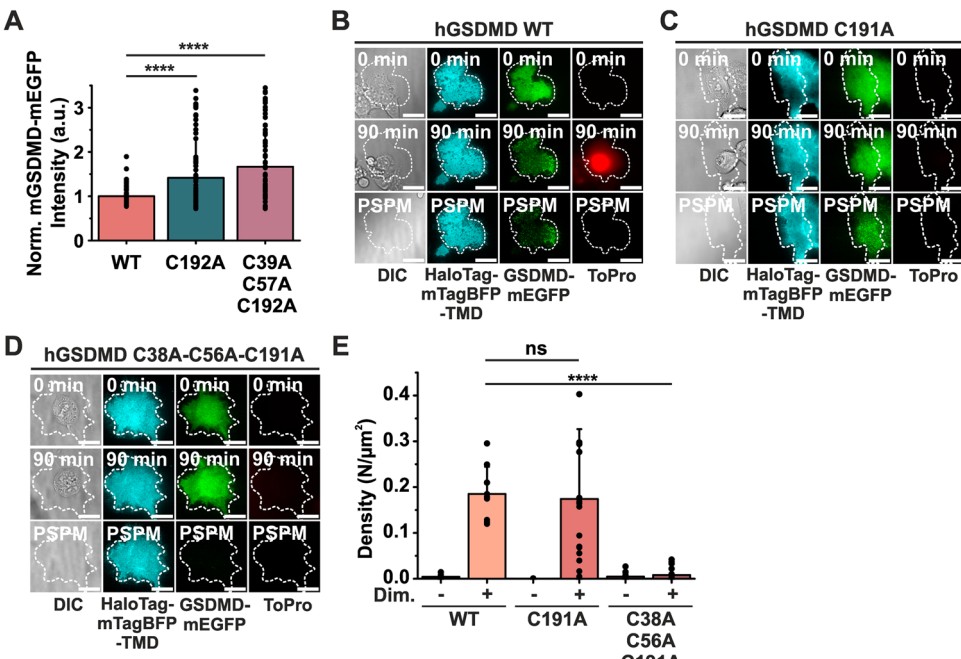

**Figure EV5. Mutation at multiple cysteines abrogates GSDMD membrane targeting in cells.**

(A) Quantification of the average expression level of stable transfected WT mGSDMD-mEGFP or transiently transfected mGSDMD-mEGFP C192A or C39A-C57A-C192A in HEK293T cells stably expressing mCaspase 1 determined by the mEGFP intensity through confocal microscopy (WT: 88 cells, 3 experiments; C192A: 93 cells, 3 experiments; C39A-C57A-C192A: 95 cells, 3 experiments) (C192A: $p = 2.13 \times 10^{-5}$; C39A-C57A-C192A: $p = 2.5 \times 10^{-10}$). Individual experimental data points are indicated as scatter plots in the graphs. (B–D) Representative TIRF images of HEK293T cells stably transfected with mCaspase 1 and transfected with hGSDMD-mEGFP WT (B), C191A (C) and C38A-C56A-C191A (D), green) and HaloTag-mTagBFP-TMD (cyan) for stable tethering on a PLL-PEG-HTL-coated surface. Images after pyroptosis induction indicated by ToPro3-Iodide staining (ToPro, red), morphological changes (DIC) and hGSDMD-mEGFP oligomers (green), and after PSPM formation. Scale bars 20 µm. (E) Average density of detected hGSDMD-mEGFP clusters in PSPMs (WT + Dimerizer: 9 cells, 2 experiments; WT - Dimerizer: 7 cells, 2 experiments; C191A + Dimerizer: 18 cells, 3 experiments; C191A - Dimerizer: 14 cells, 3 experiments; C38A-C156A-C191A + Dimerizer: 20 cells, 3 experiments; C38A-C56A- C191A - Dimerizer: 19 cells, 3 experiments) (C191A: $p = 0.839$; C38A-C56A-C191A: $p = 5.1 \times 10^{-13}$). Individual experimental data points are indicated as scatter plots in the graphs. Error bars represent SD from the different experiments. Statistics were measured by Student's t-tests with **** for $p < 0.0001$ and ns (non-significant) for $p > 0.05$.

