## [Peer Review File · The EMBO Journal]

Gasdermin D cysteine residues synergistically control its palmitoylation-mediated membrane targeting and assembly

Katia Cosentino, Eleonora Margheritis, Shirin Kappelhoff, John Danial, Nadine Gehle, Wladislaw Kohl, Rainer Kurre, and Ayelen Gonzalez Montoro

Corresponding author(s): Katia Cosentino (kacosentino@uos.de)

Review Timeline:

Submission Date:	12th Feb 24
Editorial Decision:	20th Mar 24
Revision Received:	31st May 24
Editorial Decision:	4th Jul 24
Revision Received:	15th Jul 24
Accepted:	17th Jul 24

Editor: Ioannis Papaioannou

Transaction Report:

Dear Dr. Cosentino,

Thank you for submitting your manuscript EMBOJ-2024-116948 for consideration by The EMBO Journal. It has been seen by four experts in the field, and we have received the full set of their comments, which I have already shared with you (they are included again below). I would also like to thank you for your draft point-by-point response to their comments and for your provisional revision plan, which were very helpful for us to reach a fair and balanced editorial decision on the manuscript.

The referees recognize that the study is interesting and timely, they find the data of high quality and the results intriguing. However, they also identify limitations and raise a number of concerns, and they provide detailed suggestions for further improvement and strengthening of the manuscript.

Given the referees' comments and recommendations, as well as your willingness to revise your study substantially, I would like to invite you to submit a revised version of the manuscript focusing on expanding your investigation to provide more conclusive evidence for cysteine palmitoylation, assess the effects of cysteine mutations on palmitoylation, and rule out the possibility that other cysteines in the gasdermin D N-terminal domain may affect its permeabilization activity. I would like to note that although we agree with referee #2 that more relevant cells such as macrophages would be more suitable for demonstrating the physiological relevance of the findings, such experiments will not be required for further consideration of your manuscript by The EMBO Journal.

Please also address the comments and suggestions of all four referees in a detailed point-by-point response. I should add that it is EMBO Journal policy to allow only a single round of major experimental revision, and acceptance of your manuscript will therefore depend on the completeness of your responses in this revised version. If you have any questions or comments, we can discuss further in a video call, if you like.

We generally allow three months as standard revision time (June 19, 2024). As a matter of policy, competing manuscripts published during this period will not negatively impact our assessment of the conceptual advance presented by your study. However, we request that you contact us as soon as possible upon publication of any related work, to discuss how to proceed. Should you foresee a problem in meeting this three-month deadline, please let us know in advance and we may be able to grant an extension.

Thank you for the opportunity to consider your work for publication in The EMBO Journal. I look forward to your revision.

Best regards,

Ioannis

Instructions for preparing your revised manuscript

1. When you are ready to submit the revision, please upload:

- A Word file of the manuscript text (including legends of main Figures, EV Figures and Tables). Please make sure that changes are highlighted (or "tracked") to be clearly visible.

- Individual production-quality figure files (one file per figure). When assembling your figures, please refer to our figure preparation guidelines in order to ensure proper formatting and readability in print as well as on screen:

If the data shown in a figure are obtained from n {less than or equal to} 2, please use scatter plots showing the individual data points.

i. the name of the statistical test used to generate error bars and P values

ii. the number (n) of independent experiments (please specify technical or biological replicates) underlying each data point

(discussion of statistical methodology can be reported in the Materials and Methods section, but figure legends should contain a

basic description of n, P, and the test applied)

iii. the nature of the bars and error bars (s.d., s.e.m.).

- A point-by-point response to the referees' comments, with a detailed description of the changes made (as a word file). All referees' concerns must be fully addressed and their suggestions taken on board. When preparing your letter of response to the referees' comments, please bear in mind that this will form part of the Review Process File and will therefore be available online to the community. Please note that you have the possibility to opt out of the transparent process at any stage prior to publication by letting the editorial office know (contact@embojournal.org); if you do opt out, the Review Process File link will point to the following statement: "No Review Process File is available with this article, as the authors have chosen not to make the review process public in this case.". For more details on our Transparent Editorial Process, please visit our website:

<https://www.embopress.org/page/journal/14602075/authorguide#transparentprocess>

- Expanded View (EV) files (replacing Supplementary Information) that are collapsible/expandable online. A maximum of 5 EV Figures can be typeset. EV Figures should be cited as "Figure EV1, Figure EV2" etc. in the text, and their respective legends should be included in the manuscript file after the legends of regular figures. See detailed instructions regarding Expanded View files here:

- For the figures that you do NOT wish to display as Expanded View figures, they should be bundled together with their legends in a single PDF file called "Appendix", which should start with a short Table of Contents (including page numbers). Appendix figures should be referred to in the main text as: "Appendix Figure S1, Appendix Figure S2" etc. Please see detailed instructions here: <https://www.embopress.org/page/journal/14602075/authorguide#expandedview>

- A complete author checklist, which you can download from our author guidelines

(<https://www.embopress.org/page/journal/14602075/authorguide>). Please note that the checklist will also be part of the Review Process File.

2. Please note that no statistics should be calculated and shown in Figures if $n=2$.

3. Before submitting your revision, primary datasets (and computer code, where appropriate) produced in this study need to be deposited in appropriate public databases (see <https://www.embopress.org/page/journal/14602075/authorguide#dataavailability>). The accession numbers and databases should be listed in a formal "Data availability" section (placed after Materials and Methods) that follows the model below (see also

<https://www.embopress.org/page/journal/14602075/authorguide#dataavailability>):

Data availability

- RNA-seq data: Gene Expression Omnibus GSE46843 (<https://www.ncbi.nlm.nih.gov/geo/query/acc.cgi?acc=GSE46843>)

- [data type]: [name of the resource] [accession number/identifier/doi] ([URL or identifiers.org/DATABASE:ACCESSION])

*** All links should resolve to a page where the data can be accessed. ***

*** Please remember to provide in the Data availability section of your revised manuscript reviewer passwords if the datasets are not yet public. ***

*** The Data Availability Section is restricted to new primary data that are part of this study. In case you have no data that require deposition in a public database, please state so instead of referring to the database: "Our study includes no data deposited in public repositories." under the heading "Data availability". ***

4. Please check that the title and the abstract of the manuscript are brief, yet explicit, even to non-specialists. The length of the title should not exceed 100 characters, and the abstract should be a single paragraph not exceeding 175 words.

5. Please also note our reference format: <https://www.embopress.org/page/journal/14602075/authorguide#referencesformat>.

7. Please remember: digital image enhancement is acceptable practice, as long as it accurately represents the original data and conforms to community standards. If a figure has been subjected to significant electronic manipulation, this must be noted in the figure legend or in the "Materials and Methods" section. The editors reserve the right to request original versions of figures and

the original images that were used to assemble the figure.

8. Our journal encourages inclusion of data citations in the reference list to directly cite datasets that were obtained from public databases. Data citations in the article text are distinct from normal bibliographical citations and should directly link to the database records from which the data can be accessed. In the main text, data citations are formatted as follows: "Data ref: Smith et al, 2001" or "Data ref: NCBI Sequence Read Archive PRJNA342805, 2017". In the Reference list, data citations must be labeled with "[DATASET]". A data reference must provide the database name, accession number/identifiers, and a resolvable link to the landing page from which the data can be accessed at the end of the reference. Further instructions are available at: <https://www.embopress.org/page/journal/14602075/authorguide#referencesformat>.

9. We request authors to consider both actual and perceived competing interests. Please review our policy (<https://www.embopress.org/page/journal/14602075/authorguide#conflictsofinterest>) and update your competing interests statement if necessary. Please name this section 'Disclosure and competing interests statement' and place it after the Acknowledgements section.

10. Please note that all corresponding authors are required to provide an ORCID ID upon submission of a revised manuscript (<https://orcid.org/>). Please find instructions on how to link your ORCID ID to your account in our manuscript tracking system in our Author guidelines (<https://www.embopress.org/page/journal/14602075/authorguide#authorshipguidelines>).

11. We use CRediT to specify the contributions of each author in the journal submission system. CRediT replaces the author contribution section, which should be removed from the manuscript. Please use the free text box to provide more detailed descriptions. See also guide to authors: <https://www.embopress.org/page/journal/14602075/authorguide#authorshipguidelines>.

13. We would also welcome the submission of cover suggestions or motifs to be used by our Graphics Illustrator in designing a cover.

14. Please use the link below to submit your revision:
<https://emboj.msubmit.net/cgi-bin/main.plex>

Referee #1:

The manuscript by Margheritis et al. describes molecular mechanisms of membrane binding and pore formation by gasdermin D (GSDMD). This is an important mediator of cell death, which allows execution of pyroptosis after the appropriate signals are received. GSDMD binds to plasma membrane and form pores by a poorly understood mechanism and this manuscript provide some additional insights into this process. Two aspects of GSDMS pore formation are explored, one involves initial steps of pore formation, i.e. binding and oligomerization, and the other relates to post-translational modifications of GSDMD, which may play a role in membrane engagement and oligomerization. The work is executed at a high technical level, the authors use one of their previously developed methods for characterization of plasma membranes. This is crucial for this work and is rightly used as presented in the manuscript.

The findings are intriguing. It is shown that GSDMD assembles, after binding, into dimeric and trimeric building blocks, which is a novel observation. It is also presented that S-palmitoylation is required for membrane binding, together with the presence of negatively charged lipids. Finally, it is shown that some of the GSDMD cysteines are important for this process. Nevertheless, I found the presented work not yet fully convincing. Specifically, the work related to cysteine mutagenesis would need to be extended in order to fully comprehend all the new insights that this excellent study provides. Following are the comments that I believe would need to be addressed.

Page 5: it is claimed that recombinant GSDMD appeared to be a monomer in solution, however, the extended data Fig. 1F shows it to be predominately dimeric. Please clarify or adjust the text accordingly.

Page 5: the approach that the authors used include incubation of protein with liposomes in order to bind to the membranes and oligomerize. One hour is chosen for this. Did authors attempt to do the experiment and corresponding analysis at other times (shorter and longer time scales). Please clarify why one hour incubation time is used. Would the percentage of different oligomeric assemblies be different at different timescales (i.e. as shown in Fig. 1F and G)?

Page 5 and 15: for showing the brightness of oligomers a probability density function was generated. Can this be explained in couple of sentences to non-specialists in the corresponding materials and methods section?

Page 5 and Figure 1: several panels show quantification of the amount of different oligomeric species (i.e. Fig. 1H and I). These data have very high errors, as a rule also in other figures. Please comment in the text. I think a statistical evaluation of the obtained results would also need to be performed, i.e. if different groups (oligomeric species) are significantly different from each other. Also, could individual data points be also presented in order to get a better overview of the variability of the experiments?

Page 5 and Figure 2: In Fig. 2B some minor bright dots are seen in the case of mGSDMD-mEGFP sample. Can this small percentage of bound control protein sample be quantified? Could these dots represent non-cleavable dimers from protein purification as shown in Extended Data Fig. 1?

Page 6 and Figure 2: The data in Fig. 2F show that mostly trimers or combinations of thereof are present and this looks different from data shown in Fig. 1., where dimers and trimers are present. Please comment or rephrase the corresponding section of results on page 6.

Page 7: Cysteine mutagenesis is not comprehensive. What about other three cysteines in GSDMD? A very compelling case should be made by authors why other three cysteines are not included in the study. Ideally, the alanine mutations should be done to show they do not affect permeabilization process.

Page 7: what is the quality of the recombinant proteins used in Figure 4? Could loss of activity for the triple mutant be the result of changed structure? Gels of prepared proteins should be shown in supp info, together with some other quality control data, if possible (CD spectra, melting curves from DSF, etc.).

Page 8: In addition, palmitoylation should be performed with mutant proteins in order to see whether mutations affect palmitoylation, and consequently, different steps of pore forming mechanism by using membrane data. In absence of such data it is hard to derive conclusion that the role of the three cysteines included in the study is to mediate GSDMD targeting through their palmitoylation (top of the page 8).

Page 15: How were mobile or immobile particles defined? It should be better defined in the corresponding Tracking Analysis section.

Figure 3: Palmitoylation of recombinant full-length mGSDMD (Molecular weight of 55kDa) detected by ABE is shown twice, in Fig. 3D and Extended Fig. 1H. I suggest the control protein Vac8 is shown in Fig. 3D and panel H is removed from Extended Data Fig. 2.

Figure 4: panels I and J are mentioned in the legend, please correct.

Figure 6: residues investigated in the study are shown in green, but this is not well visible on the figure.

Referee #2:

Gasdermins are a family of recently described pore-forming proteins that promote inflammation and host defence. In this study, the authors investigated the mechanisms by which gasdermin D (GSDMD) pores are assembled and targeted to the plasma membrane. This is a timely and interesting study, and I have listed some suggestions to improve the manuscript, listed below.

Major comments

1. The major limitation of this study is the lack of a physiological cellular model. Overexpression system in HEK293T cells has its merits but does not fully recapitulate the complexity of a pyroptosis-competent cell such as a macrophage. I suggest that the authors repeat most of their key findings in a physiological relevant cell type, for example by lentiviral transduction, to demonstrate that the single, double, or triple cysteine mutants are no longer able to oligomerise into higher oligomer (e.g. cross-linking assays) and insert into the plasma membrane (membrane fractionation).
2. These studies, listed in point 1, should also be complemented by IL-1b/18 secretion and LDH release.
3. I appreciate that the technical challenges of TIRF imaging, but I am concerned about the design of the GSDMD-GFP protein, for two main reasons. (1) Doesn't GFP have a high propensity to oligomerise by itself? Thus, can the authors use another fluorescence tag, such as mCherry, to demonstrate that the same results will be obtained? (2) Feng Shao recently demonstrated that lengthening or shortening the linker sequence between the pore-forming domain and the autoinhibitory domain can severely impact its cleavage by inflammatory caspases. How can the authors be sure their construct is not compromised?
4. In Figure 5, the authors demonstrate that mutating the cysteine to alanine abrogates cell death, but did not demonstrate that GSDMD cleavage is not affected. And, how much of this will be influenced by the GFP tag?

5. Other studies showed that mutating cysteine 191 can completely abrogate cell death, but this finding is inconsistent with this study (all three cysteines are required). Why is this the case?

Referee #3:

In this study, Margheritis et al. establish a model for oligomerisation, membrane insertion and pore formation by murine GSDMD based on cooperative actions of three Cys residues in the N-terminal GSDMD fragment that is generated upon caspase-mediated cleavage. The authors used state-of-the-art single molecule imaging combined with a novel technology that allows imaging of plasma membranes devoid of the cellular corpses in order to visualize in situ GSDMD pores on pyroptotic cell membranes. These analyses showed that combined actions of three Cys residues were necessary for forming GSDMD pores. While Cys192 mutants could still insert in the membrane without causing leakage, combined Cys39/57/192 mutations abolished the ability of GSDMD to insert in the membrane. The authors suggest that palmitoylation of the cysteines facilitated membrane targeting as well as oligomerisation to dimers and trimers. However, membrane-targeted higher-order oligomers were not observed with single or double Cys GSDMD mutants, suggesting that actual pore formation relies on the presence of all three of those residues.

This is a high-quality study with novel insights that will certainly advance the field. There are currently two preprints in BioRxiv (both referenced by the authors) that suggest similar palmitoylation-dependent mechanisms of GSDMD pore formation, thereby strengthening the conclusion of the current study. However, I believe the current study is nicely complementary to these preprints and therefore merits publication after addressing the one major comment below. Addressing the minor comments would be nice if achievable in a short timeframe, but are not essential.

Major comment:

1. Both BioRxiv preprints show data suggesting that GSDMD is exclusively palmitoylated at Cys192 but not on other Cys residues. In this study, the authors show redundancy between Cys192 and Cys39/57. Both the Cys192 mutant and the Cys39/57 mutant are still targeted to the membrane as dimers and trimers. This suggests that palmitoylation of Cys39/57 could compensate for the lack of Cys192 palmitoylation and vice versa, facilitating membrane targeting and di-/trimerization. The authors showed palmitoylation of the WT GSDMD using an ABE assay in Fig 3D, but they did not do this with the various single, double and triple Cys/Ala mutant GSDMD proteins they have at their disposal. Performing such assays would shed more light on how palmitoylation of various Cys residues cooperates in GSDMD function, which would also allow refining the model in Fig 6 with possible individual contributions of the various Cys residues in the pore-forming process.

Minor comments:

1. The authors show in Extended Fig 5D that the human Cys191 mutant behaves similarly as the murine Cys192 mutant. It would be nice to include also a human triple Cys/Ala mutant experiment to evaluate whether also in human GSDMD there is a cooperation between three Cys residues in order to establish a functional GSDMD pore in the plasma membrane.
2. Please include also the WT condition in Fig 4C.

Referee #4:

Activation of pore-forming protein GSDMD by innate immune inflammasome pathways causes plasma membrane rupture and induces a proinflammatory cell death known as pyroptosis. Recently, a series of biochemical and structural studies have already uncovered the structural basis for GSDMD activation and pore assembly. However, whether GSDMD targeting and perforation of lipid membranes is regulated by other yet-to-be studied biological events remains unknown. In the submitted manuscript, Margheritis, et al. take advantage of an SMI approach combined with intensity-based stoichiometry analysis to investigate the detailed pore-forming process of active GSDMD, as well as the functional roles of several cysteine residues within the GSDMD-N domain for the pore-forming activity. Their data suggest active GSDMD forms low-order oligomers as the building units for membrane pore formation, and multiple cysteine residues, namely C39/C57/C192, undergo palmitoylation that cooperatively governs GSDMD-N domain targeting membrane and assembling into pores. While the experiments provide some intriguing or potentially interesting insights into the delicate process of GSDMD pore assembly and regulation by the PTM palmitoylation, there are serious logical gaps between the data and the interpretations. Their experimental design in obtaining and characterizing GSDMD protein did not lead to generation of strong and direct evidences that can back up the conclusion. Thus, the reviewer cannot recommend to publish the manuscript in its current format.

1) My major criticism is that whether the recombinant mGSDMD-mEGFP protein expressed in *E. coli* is indeed functionally palmitoylated. The palmitoylation was largely assumed but not verified sufficiently before being applied into the SMI and stoichiometry analyses on SLBs. The authors provided ABE assay results to evidence that their sample is potentially modified.

However, this is an indirect assay to detect the palmitoylation on protein cystines. This qualitative assay can not quantify the modification ratio. Also, because of lack of necessary cystine mutations (key negative controls), one can hardly conclude that the signals in the ABE assay represent real palmitoylation occurring at the function-relevant C39/C57/C192 sites. As pointed out by the authors, bacteria have no specific palmitoyltransferases. Indeed, a recent similar study (referenced as 48) that also examined bacterially expressing GSDMD showed that this protein is short of palmitoylation. Reference 48 instead suggests that this type of PTM happens dominantly at C192 only in mammalian cells. As a result, C192A mutation in bacteria-derived GSDMD causes no detectable defects in permeabilizing reconstituted liposome membranes. These results are controversial to the data present here by the authors and should be clarified.

2) The data in fig.1 suggests that the oligomerization state of GSDMD-N domain in the SLB is concentration-dependent, in which more GSDMD-N domain induce higher oligomers. How do the authors justify the concentration of 2 nM is a reasonable one to determine the building unit of membrane-associated GSDMD-N domain? If reducing the concentration of active mGSDMD-mEGFP further, could it be possible that GSDMD-N monomer is the prominent species bound in the SLB?

3) Comparison of the data in fig.2E and 2F with that in fig. 1F and 1H suggests that 2 nM of active mGSDMD-mEGFP forms significantly more high-order oligomers in preformed SLB but only dimers and trimers in free liposomes before being transformed to SLB. This observation is strange and counterintuitive given that free liposomes should have less restrains on oligomeric assembly of the GSDMD-N domain than the preformed SLB. How to explain this unexpected observation? Are there appropriate controls needed?

4) The effects of reductants on GSDMD-N domain oligomerization in SLB and perforation in liposomes is also quite questionable. In fact, the recombinant mGSDMD-mEGFP proteins used for all the tests were purified in the buffers containing 1 mM DTT, as described in the Methods section. If the palmitoylation on GSDMD is sensitive to reductants, they should already have been removed during the purification process. Otherwise, the purified proteins should barely contain cystine palmitoylation and insensitive to the reductants used.

5) The interpretation that Cys57 and Cys39 play a redundant or compensatory role in Cys192 palmitoylation required for GSDMD-N domain oligomerization in membranes is unsubstantiated and much pre-mature. Furthermore, there is no direct evidence to show that both cystine residues could be palmitoylated. From a structural point of view, Cys192 locates in the transmembrane beta-strand and presumably contacts with the membrane lipids, therefore palmitoylation of Cys192 may increase the hydrophobicity of the transmembrane elements that can facilitate their membrane targeting and stabilization during the oligomeric assembly process. However, Cys57 and Cys39 belongs to the globular domain distant from the membrane. The cryo-EM structure of GSDMD pore has already demonstrated that both cystine residues are situated at the inter-subunit interface. Their mutations may directly cause some unappreciated interferences to subunit assembly for pore formation instead of indirectly through the palmitoylation.

Referee #1

The manuscript by Margheritis et al. describes molecular mechanisms of membrane binding and pore formation by gasdermin D (GSDMD). This is an important mediator of cell death, which allows execution of pyroptosis after the appropriate signals are received. GSDMD binds to plasma membrane and form pores by a poorly understood mechanism and this manuscript provide some additional insights into this process. Two aspects of GSDMD pore formation are explored, one involves initial steps of pore formation, i.e. binding and oligomerization, and the other relates to post-translational modifications of GSDMD, which may play a role in membrane engagement and oligomerization. The work is executed at a high technical level, the authors use one of their previously developed methods for characterization of plasma membranes. This is crucial for this work and is rightly used as presented in the manuscript.

The findings are intriguing. It is shown that GSDMD assembles, after binding, into dimeric and trimeric building blocks, which is a novel observation. It is also presented that S-palmitoylation is required for membrane binding, together with the presence of negatively charged lipids. Finally, it is shown that some of the GSDMD cysteines are important for this process. Nevertheless, I found the presented work not yet fully convincing. Specifically, the work related to cysteine mutagenesis would need to be extended in order to fully comprehend all the new insights that this excellent study provides. Following are the comments that I believe would need to be addressed.

We thank the reviewer for the critical evaluation and appreciation of our work, as well as for acknowledging its novelty.

1. Page 5: it is claimed that recombinant GSDMD appeared to be a monomer in solution, however, the extended data Fig. 1F shows it to be predominately dimeric. Please clarify or adjust the text accordingly.

We agree with the reviewer that this point was misleading. Unlike the SDS and mass photometry data (Figure EV 1B and C, respectively), which primarily show monomeric full-length GSDMD-mEGFP, the stoichiometry data in old Figure EV 1F indicated a higher proportion of dimers. Repeating the assay with a new set of data (new Figure EV 1F) still shows a high percentage of dimers, with high variability between experiments. We partially explained this discrepancy because the Halo-GFP Nanobody (Nb) used to recruit mGSDMD-mEGFP to the PLL-PEG-HTL (Figure EV 1D) might not always distribute homogeneously on the PLL-PEG surface, leading to some aggregation and resulting in mEGFP dimers. Supporting this explanation, mEGFP recruited to the surface in the same manner (Figure EV 1E) shows a percentage of dimers that can reach up to 50%. In contrast, mEGFP directly deposited on PLL-PEG-HTL (without the GFP-Nb anchor) (Figure EV 1H) results in nearly 100% monomers (Figure EV 1J). Of note, the calibration in our stoichiometry assays has been executed following the strategy presented in Figure EV 1H, thus ensuring no bias due to Nb anchoring.

Still, a certain percentage of dimeric full-length GSDMD-mEGFP indeed exists in our experiments. These dimers, as clarified in the manuscript, likely stem from purification artifacts, consistent with existing literature (Liu et al. Nature 2016). It is important to note that their concentration may vary between purification batches, as visible from the individual experiments, reported as dots in Figure EV 1F, resulting in the observed high error bars. However, it's crucial to emphasize that these dimers do not participate in the oligomerization process, as they remain uncleavable by caspase (Figure EV 1G) and have none or very limited binding to the membrane (Figure 2B).

We have better clarified this point in the revised text (p. 5).

2. Page 5: the approach that the authors used include incubation of protein with liposomes in order to bind to the membranes and oligomerize. One hour is chosen for this. Did authors attempt to do the experiment and corresponding analysis at other times (shorter and longer time scales). Please clarify why one hour incubation time is used. Would the percentage of different oligomeric assemblies be different at different timescales (i.e. as shown in Fig. 1F and G)?

We thank the reviewer for pointing this out. We determined that a 60-minute incubation time represents a fully steady-state condition, as evidenced by the calcein assay for mGSDMD WT (Appendix Figure S1A, B). Consequently, extending the time points beyond this duration would not provide additional insights. Furthermore, our calcein assay release experiments indicate a rapid oligomerization process, with full liposome permeabilization occurring within minutes of protein addition to the liposomes. However, we acknowledge the reviewer's valid point regarding shorter time intervals. To address this concern, we conducted measurements at 2 and 5 minutes incubation, spanning various GSDMD concentrations. Our data confirm the concentration dependency observed at 60 minutes and clearly highlight that the process of oligomerization is very fast (within 2 minutes) and at 5 minutes has already reached the same distribution than 60 minutes. We have added these results in Appendix Figure S1 and updated the text accordingly (p. 5). This new data provide a more comprehensive understanding of the GSDMD oligomerization process.

3. Page 5 and 15: for showing the brightness of oligomers a probability density function was generated. Can this be explained in couple of sentences to non-specialists in the corresponding materials and methods section?

We explained this better in the Methods section of the manuscript.

4. Page 5 and Figure 1: several panels show quantification of the amount of different oligomeric species (i.e. Fig. 1H and I). These data have very high errors, as a rule also in other figures. Please comment in the text. I think a statistical evaluation of the obtained results would also need to be performed, i.e. if different groups (oligomeric species) are significantly different from each other. Also, could individual data points be also presented in order to get a better overview of the variability of the experiments?

Our stoichiometry data unequivocally demonstrate the assembly of GSDMD into dimers and trimers, which serve as the fundamental building blocks of GSDMD oligomerization. However, the proportions of these species may vary between experiments. In turn, higher-order oligomers, which originate from the combination of dimers and trimers, may also exhibit variability in the proportion across experiments thus originating the high errors observed in the quantification. However, this variability does not undermine the quality and relevance of our findings. They clearly establish that oligomerization originates from these dimeric and trimeric building blocks. We emphasized this point in the revised manuscript to ensure clarity (p. 5). Furthermore, we included scatter plots showing individual experimental data points in all the bar graphs in the manuscript to offer a clearer overview of experiment-to-experiment variability.

5. Page 5 and Figure 2: In Fig. 2B some minor bright dots are seen in the case of mGSDMD-mEGFP sample. Can this small percentage of bound control protein sample be quantified? Could these dots represent non-cleavable dimers from protein purification as shown in Extended Data Fig. 1?

We appreciate the reviewer's suggestion to investigate whether the minor bright dots observed in Figure 2B represent non-cleavable dimers from protein purification. Typically, our experiments

involve analyzing hundreds to thousands of particles per experiment to ensure robust statistical analysis. However, achieving such numbers in experiments with full-length mGSDMD-mEGFP may be challenging due to the low particle density. Nevertheless, we acknowledge the importance of addressing this question and we performed the analysis as requested. Although the reduced number of detected and analyzed particles limits the robustness of these results, we observed the presence of dimers and, to a lesser extent, higher-order oligomers. However, it remains unclear whether these species arise from non-cleavable preformed dimers, as hypothesized by the reviewer, or from aggregated monomeric full-length protein. We added these results in Appendix Figure S2C and we also implemented the text accordingly (p.6).

6. Page 6 and Figure 2: The data in Fig. 2F show that mostly trimers or combinations of thereof are present and this looks different from data shown in Fig. 1., where dimers and trimers are present. Please comment or rephrase the corresponding section of results on page 6.

As mentioned in point 4, while we consistently observe both dimers and trimers, their proportions may vary significantly across experiments. However, it's important to note that despite these variations, their combination results in a similar distribution of higher-order oligomers (predominantly 5-, 6-, and 9-mers), as seen in both Figure 1I and Figure 2F. We clarified this point in the manuscript to avoid any ambiguity (p.6). Additionally, to enhance transparency and clarity in presenting our data, we included more experiments and individual experimental data points in the bar graph for both figures.

7. Page 7: Cysteine mutagenesis is not comprehensive. What about other three cysteines in GSDMD? A very compelling case should be made by authors why other three cysteines are not included in the study. Ideally, the alanine mutations should be done to show they do not affect permeabilization process.

We focused on cysteines 39, 57, and 192 because they are the only cysteines conserved in humans and mice, and Cys39 and Cys192 have been previously implicated in oligomerization. Our findings clearly show that simultaneous mutations of these cysteines are sufficient to disrupt membrane binding and affect oligomerization.

However, we thank the reviewer for raising this valid concern regarding the other three cysteines present in the N-Domain of mouse GSDMD, as has led us to perform a more systematic analysis that strongly supports our findings and improves our work. Following the reviewer's suggestion, we performed single alanine mutations of all cysteines not previously included in our study. The new data are presented in Figure EV4 and Appendix Figures S3 and S4.

Notably, C57A exhibited a similar behavior to the single mutant C39A, with reduced permeabilization and oligomerization ability compared to the WT, but still forming higher-order oligomers (Figure EV4). In contrast, mutations of cysteines 77, 122, and 265 did not affect either GSDMD ability to fully permeabilize liposomes (Appendix Figure S3) or to oligomerize (Appendix Figure S4). These new findings are now described and compared on page 8-9 of the revised manuscript.

Overall, our comprehensive analysis of cysteines in GSDMD oligomerization has led to the following key conclusions:

- 1) Cys39, 57 and 192 are the only cysteines relevant in GSDMD oligomerization.
- 2) Cys192 alone is sufficient for the formation of dimer and trimer building blocks, but requires the presence of at least one additional cysteine (Cys39 or 57) to form higher-order oligomers.

- 3) Cys39-Cys57 together play a role similar to that of Cys192 alone: their presence is sufficient to form dimers/trimers but they require additionally Cys192 to form higher-order oligomers.

These findings indicate that Cys192 and Cys39/57 have a redundant role in dimer/trimer formation, but not in higher-order oligomer formation.

These conclusions, together with our new findings on GSDMD palmitoylation (please see answer 9), has led us to a new formulated model presented in new Figure 6 and discussed on page 11 of the revised manuscript:

GSDMD oligomerization relies on a two-step mechanism mediated by cysteines: Step I (mediated either by Cys192 alone or Cys39/57) for the formation of dimers and trimers, and Step II (mediated by the cysteines not involved in Step I) for the formation of higher-order oligomers. For a more detailed discussion of the model, please see answer 9.

We believe that these new data, together with the previous stoichiometry experiments, provide compelling evidence for the key and cooperative role of cysteines 39, 57, and 192 in GSDMD oligomerization, and we hope this satisfy the reviewer's concerns.

8. Page 7: what is the quality of the recombinant proteins used in Figure 4? Could loss of activity for the triple mutant be the result of changed structure? Gels of prepared proteins should be shown in supp info, together with some other quality control data, if possible (CD spectra, melting curves from DSF, etc.).

To address the reviewer's query and provide additional transparency, we included gels of the prepared proteins in Appendix Figure S6E.

The mutant proteins utilized in Figure 4 have been previously employed in other studies (e.g. Devant et al., Cell reports 2023; Rathkey et al., Science Immunol 2018, Wang et al. JMCB 2019), ensuring their established utility and reliability. However, to ensure the robustness and reliability of our protein preparations and to satisfy the reviewer's concern, we performed protein thermal shift melting curves for all single mutants, C39A-C57A double mutant and C39A-C57A-C192A triple mutant. We could not observe any significant difference in the melting temperatures amongst them, suggesting no structural changes. We have included this analysis in the manuscript in Appendix Figure S6 A-D.

9. Page 8: In addition, palmitoylation should be performed with mutant proteins in order to see whether mutations affect palmitoylation, and consequently, different steps of pore forming mechanism by using membrane data. In absence of such data it is hard to derive conclusion that the role of the three cysteines included in the study is to mediate GSDMD targeting through their palmitoylation (top of the page 8).

We thank the reviewer for this excellent suggestion, which has led to experiments that shed new light on our findings and on the specific role of the cysteines involved in GSDMD oligomerization.

We have conducted palmitoylation assays for the mutants using the Acyl-resin-assisted capture (acyl-RAC) approach, which requires one less protein precipitation step and an alternative pull-down approach compared to the ABE assay. The results are reported in the new Figure 4B.

This assay revealed that Cys192 is the most palmitoylated cysteine in GSDMD, as palmitoylation is drastically reduced when this cysteine is mutated. We could still observe a residual faint band in the

gel, but it could not be attributed to Cys39-Cys57 (as we initially supposed) and it was not biologically relevant, as AcylRAC assay of the C39A-C57A-C192A triple mutant, which completely abolishes GSDMD membrane association (Figure 4C), showed similar residual palmitoylation.

These important findings indicate that palmitoylation is crucial for the function of Cys192 but not for Cys39/57, suggesting that other mechanisms are involved in the coordinated action of these three cysteines in GSDMD pore formation.

At the light of these novel findings and our stoichiometry results (answer 7), we propose a, in our opinion compelling, new model of how Cys39, Cys57, and Cys192 cooperate in GSDMD pore assembly, summarized below and presented in new Figure 6 and in the discussion section of the revised manuscript (p.11):

Cys39, Cys57 and Cys192 are the only cysteines required for GSDMD oligomerization.

Cys192 plays a key contribution in GSDMD oligomerization by two functions. The first is favoring the stable anchoring of GSDMD monomers at the membrane via palmitoylation, thus increasing local concentration and providing a conducive orientation to oligomerization. Without palmitoylation (mutation of Cys192), the loss of membrane anchoring weakens interactions with the lipid membrane and reduces the local GSDMD concentration, thereby preventing oligomer formation beyond dimers and trimers.

Additionally, Cys192 alone (in the C39A-C57A mutant) is sufficient to form dimers/trimers, indicating a second, equally important, role in mediating the formation of these building blocks. This function is redundant with Cys39-57, as they can also form dimers/trimers in the absence of Cys192. This suggests that any of these cysteines can mediate dimer/trimer formation in what we called “cysteine-mediated transient Interaction I” or “Step I”. In line with this, only when Cys39, Cys57, and C192 are all mutated, the protein remains monomeric, unable to progress even to dimers or trimers, the minimal units capable of membrane insertion, resulting in no membrane-inserted GSDMD at all.

However, Cys192 alone is insufficient for higher-order oligomer formation, and requires at least one additional cysteine (Cys39 or Cys57). We therefore propose that a second “cysteine-mediated transient Interaction” or “step II” is needed for higher-order oligomerization.

Despite Cys39 and Cys57 in the GSDMD-C192A mutant should be sufficient to satisfy our two-step model, likely the reduced membrane presence of GSDMD (due to lack of palmitoylation) limits its oligomerization. This suggests that Cys192 has a more potent function compared to Cys39 and Cys57 alone, as it additionally contributes to oligomerization via its palmitoylation.

Overall, our model proposes a complex mechanism where these cysteines cooperate to GSDMD oligomerization based on the combined action of Cys192 palmitoylation and Cys39, Cys57, and Cys192 -mediated oligomer formation, both novel and relevant findings.

We are confident that these explanations and the new included findings provide convincing arguments about the conclusions and the impact of our work and hope that they address this reviewer’s concerns and also satisfy the concerns of reviewers 3 and 4.

10. Page 15: How were mobile or immobile particles defined? It should be better defined in the corresponding Tracking Analysis section.

We explained this better in the methods section of the manuscript.

11. Figure 3: Palmitoylation of recombinant full-length mGSDMD (Molecular weight of 55kDa)

detected by ABE is shown twice, in Fig. 3D and Extended Fig. 1H. I suggest the control protein Vac8 is shown in Fig. 3D and panel H is removed from Extended Data Fig. 2.

We followed the reviewer's suggestion and moved control protein Vac8 in the main Figure 3D.

12. Figure 4: panels I and J are mentioned in the legend, please correct.

We corrected it.

13. Figure 6: residues investigated in the study are shown in green, but this is not well visible on the figure.

We made the investigated residues more visible in Figure 6 and we revised the proposed model.

Referee #2

Gasdermins are a family of recently described pore-forming proteins that promote inflammation and host defence. In this study, the authors investigated the mechanisms by which gasdermin D (GSDMD) pores are assembled and targeted to the plasma membrane. This is a timely and interesting study, and I have listed some suggestions to improve the manuscript, listed below.

We thank the reviewer for the appreciation of our work and for its critical evaluation and suggestions.

Major comments

1. The major limitation of this study is the lack of a physiological cellular model. Overexpression system in HEK293T cells has its merits but does not fully recapitulate the complexity of a pyroptosis-competent cell such as a macrophage. I suggest that the authors repeat most of their key findings in a physiological relevant cell type, for example by lentiviral transduction, to demonstrate that the single, double, or triple cysteine mutants are no longer able to oligomerise into higher oligomer (e.g. cross-linking assays) and insert into the plasma membrane (membrane fractionation).

2. These studies, listed in point 1, should also be complemented by IL-1b/18 secretion and LDH release.

The main focus of our study is to provide novel insight in the molecular mechanisms driving GSDMD assembly into oligomeric structures. Achieving such a level of understanding on the assembly process of GSDMD pores requires a bottom-up approach using minimalistic model systems, such as lipid bilayers, devoid of the complexity of a cell membrane environment.

We understand the relevance of translating our studies of GSDMD mutants into the more physiological context of the cells and for this we have complemented our investigation in artificial lipid bilayers with experiments in HEK cells, confirming our findings that the investigated single and double, but not triple, cysteine mutants still bind the membrane. To further corroborate our data, we have extended our analysis to the human C38A-C56A-C191A triple mutant (Figure EV5 - please see also answer 2 to reviewer 3).

While we recognize that macrophages reflect a more physiologically relevant cell type for functional studies of pyroptosis, other labs have already shown that the mutants investigated in this study affect the oligomerization and functionality (i.e. permeabilization or LDH release) of GSDMD in

macrophages or THP1 cells (Devant et al., Cell reports 2023; Rathkey et al., Science Immunol 2018, Wang et al. JMCB 2019).

Recapitulating our mechanistic findings in macrophages would require considerable time and resources that exceed the given revision deadline. After consulting with the editor, we have decided to focus on maintaining a short revision timeframe, especially considering the recent publications on GSDMD palmitoylation that emerged during our manuscript's revision. Consequently, we are unable to provide results in this specific direction at this time. We hope the reviewer understands our constraints and finds the integrations we have addressed in the other points satisfactory.

3. I appreciate that the technical challenges of TIRF imaging, but I am concerned about the design of the GSDMD-GFP protein, for two main reasons. (1) Doesn't GFP have a high propensity to oligomerise by itself? Thus, can the authors use another fluorescence tag, such as mCherry, to demonstrate that the same results will be obtained? (2) Feng Shao recently demonstrated that lengthening or shortening the linker sequence between the pore-forming domain and the autoinhibitory domain can severely impact its cleavage by inflammatory caspases. How can the authors be sure their construct is not compromised?

All of our experiments use monomeric enhanced GFP (mEGFP), which has been engineered specifically to be monomeric. Furthermore, this monomeric nature of mEGFP was validated and confirmed in Figure EV1. Additionally, we opted for mEGFP over mCherry due to its superior stability.

Regarding the second concern, our decision to insert mEGFP in the flexible linker region between the N- and C-domains of GSDMD was based on previous work demonstrating that tag insertion in this area does not interfere with cleavage and function of GSDMD (Rathkey et al., JBC 2017). This construct has been successfully utilized in other studies, (e.g. Rathkey et al., Science Immunology 2018; Kondolf et al., JBC 2023; Miao et al., Immunity 2023). Furthermore, for each purification batch, we rigorously test the activity of GSDMD-mEGFP using calcein release assays, which is similar to GSDMD, demonstrating that the presence of GFP does not impair the pore-forming activity of GSDMD. To clarify this better, we have now included data comparing GSDMD-GFP and GSDMD activity using calcein release assays in Appendix Figure S1B of the manuscript. Additionally, we have provided Western blot data comparing GSDMD-GFP (WT and mutants) upon caspase treatment to demonstrate that the cleavage and extent of cleavage in the GSDMD-GFP constructs are not compromised (Appendix Figure S6F). This information has been included in the Methods section of the revised manuscript.

4. In Figure 5, the authors demonstrate that mutating the cysteine to alanine abrogates cell death, but did not demonstrate that GSDMD cleavage is not affected. And, how much of this will be influenced by the GFP tag?

To assess the reviewer's concern about the preserved ability of the different cysteine mutants to be cleaved by caspases, please refer to answer 3 and Appendix Figures S1B and S6F where we demonstrate that the different mutations and the presence of the GFP do not affect the ability of recombinant GSDMD to be cleaved by caspases.

In addition, we have conducted Western blot analysis using samples from control and pyroptotic cells expressing WT GSDMD, WT GSDMD with the GFP tag, as well as the C192A and C39A-C57A-C192A mutants. This analysis shows that the cysteine-to-alanine mutations do not affect GSDMD cleavage in cells and that the GFP tag does not interfere with this process (Appendix Figure S5B).

We have included this information in the revised manuscript (p.9).

5. Other studies showed that mutating cysteine 191 can completely abrogate cell death, but this finding is inconsistent with this study (all three cysteines are required). Why is this the case?

We would like to point out that our work is indeed in line with previous studies, as we also demonstrate that permeabilization and cell death can be abrogated by single mutation at Cys192, as shown in Figure 5 and Figure EV5.

The novelty of our findings lies in the recognition that, despite its inability to permeabilize membranes, the C192A mutant still retains the ability to associate to them. Importantly, the abrogation of membrane insertion (and not of permeabilization, which is already compromised in the single C192A mutant) occurs only when all the three cysteines, Cys 39, 57 and 192, are mutated. We can explain these observations at the light of our stoichiometry data. Mutation of Cys192 prevents the formation of higher-order oligomers, which are likely the functional oligomers inducing pyroptosis. However it does not prevent the formation of dimers and trimers, the minimal oligomeric units able to insert the membrane, thus justifying C192A retained ability to associate to membranes. Only the simultaneous mutations of all three cysteines abrogates the formation of dimers/trimers, thus completely abolishing GSDMD membrane association. For a more detailed explanation, we remind to answer 9 to reviewer 1.

In conclusion, our findings provide important novel insights into the mechanism by which these cysteines cooperate in GSDMD targeting and oligomerization leading to membrane permeabilization.

We hope that this explanation clarifies the concern raised by the reviewer.

Referee #3

In this study, Margheritis et al. establish a model for oligomerisation, membrane insertion and pore formation by murine GSDMD based on cooperative actions of three Cys residues in the N-terminal GSDMD fragment that is generated upon caspase-mediated cleavage. The authors used state-of-the-art single molecule imaging combined with a novel technology that allows imaging of plasma membranes devoid of the cellular corpses in order to visualize in situ GSDMD pores on pyroptotic cell membranes. These analyses showed that combined actions of three Cys residues were necessary for forming GSDMD pores. While Cys192 mutants could still insert in the membrane without causing leakage, combined Cys39/57/192 mutations abolished the ability of GSDMD to insert in the membrane. The authors suggest that palmitoylation of the cysteines facilitated membrane targeting as well as oligomerisation to dimers and trimers. However, membrane-targeted higher-order oligomers were not observed with single or double Cys GSDMD mutants, suggesting that actual pore formation relies on the presence of all three of those residues.

This is a high-quality study with novel insights that will certainly advance the field. There are currently two preprints in BioRxiv (both referenced by the authors) that suggest similar palmitoylation-dependent mechanisms of GSDMD pore formation, thereby strengthening the conclusion of the current study. However, I believe the current study is nicely complementary to these preprints and therefore merits publication after addressing the one major comment below. Addressing the minor comments would be nice if achievable in a short timeframe, but are not essential.

We thank the reviewer for the positive evaluation of our work, as well as for acknowledging the novelty of our study.

Major comment:

1. Both BioRxiv preprints show data suggesting that GSDMD is exclusively palmitoylated at Cys192 but not on other Cys residues. In this study, the authors show redundancy between Cys192 and Cys39/57. Both the Cys192 mutant and the Cys39/57 mutant are still targeted to the membrane as dimers and trimers. This suggests that palmitoylation of Cys39/57 could compensate for the lack of Cys192 palmitoylation and vice versa, facilitating membrane targeting and di-/trimerization. The authors showed palmitoylation of the WT GSDMD using an ABE assay in Fig 3D, but they did not do this with the various single, double and triple Cys/Ala mutant GSDMD proteins they have at their disposal. Performing such assays would shed more light on how palmitoylation of various Cys residues cooperates in GSDMD function, which would also allow refining the model in Fig 6 with possible individual contributions of the various Cys residues in the pore-forming process.

We greatly appreciate the valuable comment of the reviewer regarding the importance of performing palmitoylation assay on the different GSDMD mutants to accurately infer the role of the individual investigated cysteines to GSDMD membrane binding and function through palmitoylation.

Following the reviewer's suggestion, we have performed palmitoylation assays for the mutants. These experiments presented in new Figure 4B, have indeed disclosed that Cys192 is the prevalent palmitoylated cysteine, in line with other studies published during the revision of our manuscript (Zhang et al., nature cell biology 2024; Balasubramanian et al., Sci. Immunol. 2024; Du et al., Nature 2024).

As anticipated by the reviewer, these findings shed new light on the specific contribution of these three cysteines to GSDMD function and have allowed us to refine our proposed model on their cooperative mechanism in GSDMD pore assembly, which is presented in new Figure 6 and in the discussion section of the revised manuscript.

Overall, our new model proposes a complex mechanism where these cysteines cooperate to GSDMD oligomerization by different modes: Cys 192 via the dual function of stabilizing GSDMD at the membrane via palmitoylation-anchoring and providing transient interactions that mediate oligomer formation. This second function is shared with the two other cysteines Cys39 and Cys57.

For a detailed description and the rationale behind our new model, please refer to answer 9 of reviewer 1.

We believe that our new findings and proposed model offer compelling support for the conclusions of our work and hope they address the reviewer's concerns effectively.

Minor comments:

1. The authors show in Extended Fig 5D that the human Cys191 mutant behaves similarly as the murine Cys192 mutant. It would be nice to include also a human triple Cys/Ala mutant experiment to evaluate whether also in human GSDMD there is a cooperation between three Cys residues in order to establish a functional GSDMD pore in the plasma membrane.

Following the reviewer's suggestion, we included the investigation of human triple C38A/C56A/C191A GSDMD mutant in our analysis (Figure EV5). As for the murine triple mutant, we could not observe any cell death and membrane association, confirming that these three cysteines are indeed all required to allow pore formation for human GSDMD.

2. Please include also the WT condition in Fig 4C.

We included the WT condition in Figure 4D as requested.

Referee #4

Activation of pore-forming protein GSDMD by innate immune inflammasome pathways causes plasma membrane rupture and induces a proinflammatory cell death known as pyroptosis. Recently, a series of biochemical and structural studies have already uncovered the structural basis for GSDMD activation and pore assembly. However, whether GSDMD targeting and perforation of lipid membranes is regulated by other yet-to-be studied biological events remains unknown. In the submitted manuscript, Margheritis, et al. take advantage of an SMI approach combined with intensity-based stoichiometry analysis to investigate the detailed pore-forming process of active GSDMD, as well as the functional roles of several cystine residues within the GSDMD-N domain for the pore-forming activity. Their data suggest active GSDMD forms low-order oligomers as the building units for membrane pore formation, and multiple cystine residues, namely C39/C57/C192, undergo palmitoylation that cooperatively governs GSDMD-N domain targeting membrane and assembling into pores. While the experiments provide some intriguing or potentially interesting insights into the delicate process of GSDMD pore assembly and regulation by the PTM palmitoylation, there are serious logical gaps between the data and the interpretations. Their experimental design in obtaining and characterizing GSDMD protein did not lead to generation of strong and direct evidences that can back up the conclusion. Thus, the reviewer cannot recommend to publish the manuscript in its current format.

We thank the reviewer for acknowledging the interest of our findings and their potential insights into the so far poorly understood mechanism of GSDMD pore assembly and regulation in pyroptosis.

1) My major criticism is that whether the recombinant mGSDMD-mEGFP protein expressed in *E. coli* is indeed functionally palmitoylated. The palmitoylation was largely assumed but not verified sufficiently before being applied into the SMI and stoichiometry analyses on SLBs. The authors provided ABE assay results to evidence that their sample is potentially modified. However, this is an indirect assay to detect the palmitoylation on protein cystines. This qualitative assay can not quantify the modification ratio. Also, because of lack of necessary cystine mutations (key negative controls), one can hardly conclude that the signals in the ABE assay represent real palmitoylation occurring at the function-relevant C39/C57/C192 sites. As pointed out by the authors, bacteria have no specific palmitoyltransferases. Indeed, a recent similar study (referenced as 48) that also examined bacterially expressing GSDMD showed that this protein is short of palmitoylation. Reference 48 instead suggests that this type of PTM happens dominantly at C192 only in mammalian cells. As a result, C192A mutation in bacteria-derived GSDMD causes no detectable defects in permeabilizing reconstituted liposome membranes. These results are controversial to the data present here by the authors and should be clarified.

We thank the reviewer for this comment, which allows us to clarify this indeed relevant aspect of our work.

First, referring to the comparison made by the reviewer of our results with the ones in reference 48, we would like to point out that there is currently a more recent version of the same work published during the revision of our manuscript (Du et al., Nature 2024), in which the authors have revised their previous observations and instead provide evidence supporting our observations. Specifically, they

also show that: 1) recombinant GSDMD is indeed palmitoylated, albeit to a lesser extent compared to GSDMD expressed in mammalian cells proficient of palmitoyltransferases; 2) bacterial expressed C192A GSDMD is defective in permeabilizing liposomes compared to the bacterial WT GSDMD (see figure 1I of Du et al. 2024 Nature). Both these results are consistent with our findings and strengthen them.

That said, we agree with the reviewer that the ABE assay, although routinely used to identify palmitoylated proteins, is an indirect method for detecting palmitoylation and that it may not quantify the modification ratio accurately in the absence of appropriate negative controls, which in our case, are the GSDMD cysteine mutants.

To address these reviewer's concern and confirm that the detected signal in the ABE assay corresponds to palmitoylation at the functional relevant cysteines, we have now performed palmitoylation assay of the cysteine mutants. For this, we have used acyl-resin-assisted capture (acyl-RAC) assays that compared to ABE assay, requires one less protein precipitation step and a different pull-down approach. By performing this assay on purified cleaved protein for GSDMD-GFP WT and mutants, we have indeed identified Cys192 as the only palmitoylated cysteine (see new Figure 4B and also answer 9 to reviewer 1). These new findings on the one hand validate that GSDMD is indeed palmitoylated in bacteria, on the other hand have revealed the unique role of Cys192 in GSDMD palmitoylation, leading us to reformulate our initial model on how Cys39, 57 and 192 cooperate to GSDMD function.

For a detailed description of our new model, please refer to answer 9 of reviewer 1.

We hope that these new findings and explanations will satisfy the reviewer's concerns.

2) The data in fig.1 suggests that the oligomerization state of GSDMD-N domain in the SLB is concentration-dependent, in which more GSDMD-N domain induce higher oligomers. How do the authors justify the concentration of 2 nM is a reasonable one to determine the building unit of membrane-associated GSDMD-N domain? If reducing the concentration of active mGSDMD-mEGFP further, could it be possible that GSDMD-N monomer is the prominent species bound in the SLB?

The reviewer has a point here, and we acknowledge the importance of raising it. We originally performed experiments at lower protein concentration and finally selected a concentration of 2 nM to ensure to operate within the single-molecule regime while still having sufficient particles for robust statistical analysis.

In our experiments at lower concentrations, including 0.5 nM and 1 nM, we did not observe significant differences in the oligomerization state of GSDMD compared to the 2nM concentration. Specifically, reducing the concentration did not lead to a predominant presence of GSDMD-N monomers bound to the SLB. We have included these additional data in the Appendix Figure S2 A, B of the manuscript and revised the text on page 6.

3) Comparison of the data in fig.2E and 2F with that in fig. 1F and 1H suggests that 2 nM of active mGSDMD-mEGFP forms significantly more high-order oligomers in preformed SLB but only dimers and trimers in free liposomes before being transformed to SLB. This observation is strange and counterintuitive given that free liposomes should have less restrains on oligomeric assembly of the GSDMD-N domain than the preformed SLB. How to explain this unexpected observation? Are there appropriate controls needed?

The observation of the reviewer is correct that using the same GSDMD concentration in liposomes and on SLB we obtain different oligomeric distributions in the two experiments. The discrepancy in

oligomeric distribution arises from the difference in lipid/protein ratio between the experiments in preformed SLBs and free liposomes. During SLB preparation, excess liposomes that do not fuse with the glass support to form a flat bilayer are washed away. As a result, the total number of lipids is reduced, leading to a lower lipid/protein ratio (or in other words, more protein available per lipid molecule). To illustrate this difference, we estimated the lipid/protein ratio for GSDMD incubated with LUVs or directly added to SLB. Our calculations indicate a lipid/protein ratio of $\sim 5 \times 10^{16}$ for 2 nM GSDMD and $\sim 0.5 \times 10^{14}$ for 20 nM GSDMD incubated with LUVs, while for GSDMD directly added to SLB, the lipid/protein ratio is $\sim 3 \times 10^{14}$.

We have now included this information in the Methods section of the manuscript.

4) The effects of reductants on GSDMD-N domain oligomerization in SLB and perforation in liposomes is also quite questionable. In fact, the recombinant mGSDMD-mEGFP proteins used for all the tests were purified in the buffers containing 1 mM DTT, as described in the Methods section. If the palmitoylation on GSDMD is sensitive to reductants, they should already have been removed during the purification process. Otherwise, the purified proteins should barely contain cystine palmitoylation and insensitive to the reductants used.

It is important to note that the DTT to protein ratio differs significantly between the purification process and the experiments involving SLB and liposomes. In our single-molecule experiments on SLB, protein concentrations are in the range of 2 nM and we have demonstrated that 1 mM DTT efficiently prevents membrane binding, suggesting removal of palmitoylation acyl chains and/or reduction of cysteines involved in the transient formation of intermediate precursors of the assembly units. In the liposome experiments where protein concentration is in the range 0.5-150 nM, we tested that DTT concentration needs to be above 200 mM to appreciate an inhibitory effect on the permeabilization activity of GSDMD (see Figure EV2G). During purification, we use the amount of 1 mM of DTT for protein concentrations in the range of tens of μ M. This DTT/protein ratio has the only purpose to maintain protein stability, however based on our calcein assays experiments, this ratio is insufficient to remove palmitoylation. We clarified this aspect in the Methods section of the manuscript.

5) The interpretation that Cys57 and Cys39 play a redundant or compensatory role in Cys192 palmitoylation required for GSDMD-N domain oligomerization in membranes is unsubstantiated and much pre-mature. Furthermore, there is no direct evidence to show that both cystine residues could be palmitoylated. From a structural point of view, Cys192 locates in the transmembrane beta-strand and presumably contacts with the membrane lipids, therefore palmitoylation of Cys192 may increase the hydrophobicity of the transmembrane elements that can facilitate their membrane targeting and stabilization during the oligomeric assembly process. However, Cys57 and Cys39 belongs to the globular domain distant from the membrane. The cryo-EM structure of GSDMD pore has already demonstrated that both cystine residues are situated at the inter-subunit interface. Their mutations may directly cause some unappreciated interferences to subunit assembly for pore formation instead of indirectly through the palmitoylation.

We thank the reviewer for this valuable comment, which has led to the palmitoylation experiments of the cysteine mutants described in answer 1 showing that C192 is the predominant site of palmitoylation, as well as to revise our initial model, which we have described in details in answer 9 to reviewer 1.

At the light of our new findings, we agree with the reviewer that mutations of Cys39 and Cys57 interfere more with the assembly rather than with GSDMD membrane targeting, which is instead

mediated by Cys192 palmitoylation, more likely in synergy with other residues interacting with negatively charged lipids.

To further dissect the specific roles of Cys39 and Cys57 we have extended our calcein and stoichiometry analysis to the single C57A mutant, providing further insight into the role of this cysteine in oligomerization. Both calcein release assay and stoichiometry analysis, reveal a similar behavior for these two cysteines, showing a reduced, but not impaired, ability to permeabilize liposomes and form high-order oligomers (Figure EV4).

These findings suggest the individual contribution of these cysteines to the oligomerization process, more likely by providing what we called in our new model a “cysteine-mediated transient interaction” for oligomer formation. While these cysteines do not form disulfide bonds in crystal and cryo-EM structures, in line with the reviewer’s suggestion, we propose that they may participate in some unappreciated transient thiol-mediated interactions during oligomerization.

We have revised our discussion in the manuscript accordingly.

We hope that our new findings and conclusions will satisfactorily address the reviewer’s concerns.

Dear Katia,

Thank you again for the submission of your revised manuscript to The EMBO Journal and your patience during peer review. We have now received the comments of three of the original referees that previously assessed the first version of your manuscript (you can find their comments below). As you will see, the referees acknowledge that the manuscript has been significantly improved and that the majority of their previously raised concerns have been adequately addressed. The referees now support publication of this work in The EMBO Journal pending a minor revision to address their few remaining minor requests for improvement of data presentation and clarification. Please include in your resubmission a point-by-point response to their comments, describing in detail any changes. You are also kindly requested to make sure that all relevant papers that have been published during revision and re-review of this work are properly cited in the final version of your manuscript.

From the editorial side, there are also a few changes and corrections that we need from you before we can proceed with acceptance of the manuscript:

- Please remove figures from the final version of the main manuscript file. The figures should only be uploaded to our manuscript handling system separately as individual files. Their legends should remain in the manuscript, following the list of references (and with the headings "Figure legends" and "EV Figure legends", as appropriate).

- Please note that you can list no more than 5 keywords after the Abstract (you currently have 7).

- Please update the reference format: the citations should be in alphabetical order (not numbered), with no more than 10 co-authors listed (please use "et al." after the names of the first 10 co-authors for publications with more than 10 co-authors). For more information on our reference format please visit:

<https://www.embopress.org/page/journal/14602075/authorguide#referencesformat>.

- The Materials and Methods need to be described in the manuscript using our "Structured Methods" format, which is now required for all research articles. According to this format, the Materials and Methods section includes a "Reagents and Tools Table" -listing key reagents, experimental models, software and relevant equipment and including their sources and relevant identifiers- followed by a "Methods and Protocols" section describing the methods using a step-by-step protocol format. The aim is to facilitate adoption of the methodologies across labs. More information on this format as well as a template (.docx) for the "Reagents and Tools Table" can be found in our author guide:

<https://www.embopress.org/page/journal/14602075/authorguide#structuredmethods>.

- Please note that the previously requested Source Data for Figure panels 1C, 1D, 2B, 3A, 3D, 4C, 5A, 5C, 5E must be made permanently accessible, either by uploading them to our manuscript handling system according to the instructions you have previously received from our Source Data coordinator (in this case please state in your Data availability statement that: "Our study includes no data deposited in public repositories."), or by depositing them in a public repository where they will be publicly and permanently accessible (in this case, please provide the access information, including the repository, identifier, and specific URL, in your Data availability statement).

- Please change the heading "Declaration of interests" to "Disclosure and competing interests statement".

- The author contributions statement should be removed from the manuscript file. Instead, we now use CRediT to specify the contributions of each author in the journal submission system. Please feel free to use the free text box to provide more detailed descriptions during submission. See also our guide to authors for more information:

<https://www.embopress.org/page/journal/14602075/authorguide#authorshipmentguidelines>.

- Please include the manuscript title on the first page of your Appendix PDF file.

- Please note that EMBO press papers are accompanied online by:

A) a short (2 sentences) summary of the findings and their significance,

B) 2-5 short bullet points highlighting the key results, and

C) a synopsis image in .jpg or .png format that is exactly 550 pixels wide and 300-600 pixels high (the height is variable). Please note that the text needs to be legible at the final size. Please upload this information along with your revised manuscript (the text for A and B should be provided in a separate Word file).

- Please improve the presentation of the blots shown in Figure 7B and Appendix Figure S6 E & F: no "splice sites" can be shown without proper explanation in the respective legends; instead, lanes from different (parts of the) blots are recommended to be clearly shown as different blots, for example in different boxes.

- Please note that the exact p values are not provided in the legends of Figures 1e; 2c-d; 3b; 4d-e; 5f; EV 3b-c; EV 5a, e.

- Please note that the box plots need to be defined in terms of minima, maxima, centre, bounds of box and whiskers, and percentile in the legends of Figures 1e; 2c-d; 3b; 4e; EV 3b; EV 4d.
- Please note that information related to "n" is missing in the legend of Figure 3b.
- Please note that n=2 in Figure EV 2g. When n=2, the individual points should be shown and no statistics can be calculated/shown.
- Please note that the error bars are not defined in the legends of Figures EV 5a, e.
- Please note that the measure of center for the error bars needs to be defined in the legends of Figures 1h-i; 5f.
- Please rename your movie file "Movie EV1" and update its corresponding callout in the manuscript accordingly. Its legend should be removed from the manuscript file and instead zipped together with the movie file (in a Word or text file).

Please also note that as part of the EMBO publications' Transparent Editorial Process, The EMBO Journal publishes online a Peer Review File along with each accepted manuscript. This File will be published in conjunction with your paper and will include the referee reports, your point-by-point response and all pertinent correspondence relating to the manuscript. You can opt out of this by letting the editorial office know (contact@embojournal.org). If you do opt out, the Peer Review File link will point to the following statement: "No Peer Review File is available with this article, as the authors have chosen not to make the review process public in this case."

We look forward to seeing a final version of your manuscript as soon as possible. Please use this link to submit your revision:
<https://emboj.msubmit.net/cgi-bin/main.plex>

Best regards,

Ioannis

Referee #1:

The revised manuscript by Margheritis et al. is a much-improved version, where authors have made a lot of changes to the manuscript in order to satisfactorily address reviewers comments. I only have the following minor comments.

Appendix Figure S6: Panel D shows three different thermal shift experiments. It is claimed that error bars correspond to the SD from the different experiments, however, no error bars are shown in this figure. Variability is quite high for some of the samples (i.e. wild-type, C122A...). Better graph would be to show averages+SD and also statistical analysis should be performed. I do not think that there will be insignificant differences in melting temperatures as is claimed in the response to reviewers comments and in the materials and methods section (last paragraph of the Protein purification section).

How was the gel shown in Appendix Fig S6 stained? The gel shows a lot of additional bands. Why are protein preparations not better, if two steps were used in the purification procedure? These other protein bands could affect protein characterization presented in appendix fig. S6 (i.e. stability).

Please check labels (left, right) in the legend to Figure 4B. Also, the input region of the blot should be clearly labeled for different proteins used in the assay.

Referee #2:

Regarding the concerns previously raised by Reviewer 2:

The authors have addressed most of my concerns appropriately, and I empathise that the revision needs to be completed in a reasonable time, especially after the series of GSDMD palmitoylation papers that were published recently.

Although cleavage of recombinant GSDMD (made in E coli) are not affected by the GFP tag or any of the cysteine mutations (Fig. S6F), it is a pity that some of the key findings are not recapitulated in a physiologically relevant cell type, as it appears that caspase cleavage of GSDMD C39A/57A/192A is indeed impaired compared to GSDMD 192A when these proteins are expressed in mammalian cells (Fig S5B).

Regarding the concerns previously raised by Reviewer 4:

All of the concerns have been addressed appropriately.

Referee #3:

In this revised version of the manuscript, the authors have addressed all of my previous queries satisfactorily, which resulted in a more refined model of how GSDMD palmitoylation controls its pore-forming function. Also given the recently published articles presenting supporting observations I therefore recommend publication of this nice work.

Referee #1:

The revised manuscript by Margheritis et al. is a much-improved version, where authors have made a lot of changes to the manuscript in order to satisfactorily address reviewers comments. I only have the following minor comments.

We thank the reviewer for the positive evaluation of our revised manuscript.

Appendix Figure S6: Panel D shows three different thermal shift experiments. It is claimed that error bars correspond to the SD from the different experiments, however, no error bars are shown in this figure. Variability is quite high for some of the samples (i.e. wild-type, C122A...). Better graph would be to show averages \pm SD and also statistical analysis should be performed. I do not think that there will be insignificant differences in melting temperatures as is claimed in the response to reviewers comments and in the materials and methods section (last paragraph of the Protein purification section).

We apologize for the error in the legend of Appendix Figure S6. The legend indeed referred to panels B and C, where the lines in the graph represent the average values, and the colored areas depict the data variability as standard deviations (SD). We have corrected the text accordingly.

Following the reviewer's suggestion, we have updated the graph in panel D. It now shows the average of the three measurements, with error bars representing the SD. Additionally, we have performed a statistical analysis confirming our initial statement that the melting temperatures plotted in panel D are not significantly different ($p > 0.5$).

How was the gel shown in Appendix Fig S6 stained? The gel shows a lot of additional bands. Why are protein preparations not better, if two steps were used in the purification procedure? These other protein bands could affect protein characterization presented in appendix fig. S6 (i.e. stability).

To address the reviewer's concern, we repeated the SDS-PAGE using another batch of purified proteins and destained the gel overnight for better visibility. The new gel is included in the updated Figure S6E.

Unlike the WT protein (with no GFP labeling), which consistently shows a high degree of purity in every fraction across different purification experiments, the purification of the GFP-labeled proteins occasionally results in some residual bands. However, it is important to note that for each purification experiment, all obtained fractions were carefully checked by SDS-PAGE, and only those showing a high level of purity were selected for stoichiometry experiments.

Furthermore, while some residual bands could still be present despite using both affinity and size exclusion chromatography, the melting temperatures of the GSDMD-GFP variants are not significantly different from those of the WT, non-labeled protein, indicating that these bands do not affect protein stability (see the previous answer).

Please check labels (left, right) in the legend to Figure 4B. Also, the input region of the blot should be clearly labeled for different proteins used in the assay.

We thank the reviewer for pointing this out. We corrected the labels in the legend of Figure 4B and implemented the presentation of the input of the blot.

Referee #2:

The authors have addressed most of my concerns appropriately, and I empathise that the revision needs to be completed in a reasonable time, especially after the series of GSDMD palmitoylation papers that were published recently.

We thank the reviewer for the understanding in allowing us to provide a timely revision and for the appreciation of our revised work.

Although cleavage of recombinant GSDMD (made in E coli) are not affected by the GFP tag or any of the cysteine mutations (Fig. S6F), it is a pity that some of the key findings are not recapitulated in a physiologically relevant cell type, as it appears that caspase cleavage of GSDMD C39A/57A/192A is indeed impaired compared to GSDMD 192A when these proteins are expressed in mammalian cells (Fig S5B).

We agree with the reviewer that it is still of high interest to recapitulate the key findings of this work in a more physiologically relevant cell type, and we remain open to exploring this option in future research.

Regarding the results presented in Appendix Figure S5B, we would like to emphasize that the immunoblot shown is representative of multiple experiments and is subject to expression and loading variability. However, it clearly demonstrates that the triple mutant can be cleaved by caspase (no band corresponding to the FL protein was detected after caspase activation), thereby excluding the possibility that the impairment in membrane insertion is due to defective activation of the protein.

Regarding the concerns previously raised by Reviewer 4:

All of the concerns have been addressed appropriately.

Referee #3:

In this revised version of the manuscript, the authors have addressed all of my previous queries satisfactorily, which resulted in a more refined model of how GSDMD palmitoylation controls its pore-forming function. Also given the recently published articles presenting supporting observations I therefore recommend publication of this nice work.

Additional comments from the Editor

All editorial and formatting issues were resolved by the authors.

Dear Katia,

Congratulations on an excellent manuscript! I am very pleased to inform you that it has been accepted for publication in The EMBO Journal. Many thanks for your thorough responses to the referee concerns and for addressing all editorial requests.

If you have any questions, please do not hesitate to contact the Editorial Office. Thank you for your contribution to The EMBO Journal. Working with you has been a pleasure!

Best wishes,

Ioannis
